# TRANSDUCTIVE LEARNING FOR OUT-OF-DISTRIBUTION MOLECULAR PROPERTY PREDICTION

## ABSTRACT

Predicting molecular properties outside the training data distribution (Out-of-Distribution, OOD) is critical for accelerating drug discovery. This task requires models to extrapolate beyond known property ranges and generalize to novel chemical structures—a common failure point for standard machine learning models in realistic drug discovery scenarios. While transductive analogical reasoning shows promise, prior methods are often constrained by fixed descriptors and single-anchor comparisons. To overcome these limitations, we introduce Multi-Anchor Latent Transduction (MALT) framework, which operates directly within a learned latent space. MALT can leverage embeddings from any powerful, pre-trained molecular encoder to select multiple relevant analogues of query molecule. It then integrates the query and anchor embeddings to generate a final prediction. On rigorous OOD benchmarks targeting shifts in both property values and chemical features, MALT consistently improves generalization over standard inductive baselines. Notably, our framework also matches or surpasses the in-distribution performance of these base models. These findings establish multi-anchor transduction in latent space as an effective strategy to augment existing molecular encoders, enabling robust and extrapolative predictions needed to solve challenging discovery tasks.

## 1 INTRODUCTION

Machine learning(ML), particularly deep learning, holds immense promise for accelerating scientific discovery in drug development and materials science by learning complex structure-property relationships from data (1; 2). However, a critical limitation hinders their reliable deployment: their frequent inability to generalize to out-of-distribution (OOD) data. This weakness stems from the violation of the standard IID assumption, which causes dramatic performance drops and overconfident incorrect predictions (3). In practice, models inevitably encounter molecules with novel scaffolds or different property ranges (2; 4; 5), creating distribution shifts that generic OOD techniques often fail to address for structured molecular data (4; 6; 7).

In practical drug discovery, these OOD challenges manifest in two crucial ways. First is **covariate shift**, a major barrier as pharmaceutical companies often work with proprietary compounds built on specific chemical scaffolds absent from public training data (8); this demands model robustness to novel structures (4; 6). Another common scenario in the pharmaceutical industry is **label shift**, where models are required to extrapolate beyond the observed range of property values in order to optimize the activities of lead compounds or identify potential hazards and toxicity beyond the range of training data (3; 9). Standard inductive models struggle with both challenges, often failing on even more difficult phenomena like activity cliffs (10), where minor structural changes cause large potency differences.

Transductive learning offers a complementary paradigm better suited for OOD challenges, particularly extrapolation (11; 12). It makes predictions based on analogical reasoning between a query and known training examples (9). However, existing transductive models are often limited by: (1) relying on a single "best" anchor, which is brittle if no perfect analogy exists, and (2) performing reasoning in a fixed descriptor space, potentially missing deeper similarities captured by learned representations.

To address these limitations and enhance OOD generalization for *both* covariate and label shifts, we propose a flexible transductive learning framework. Our approach operates within learned latent spaces and integrates seamlessly with **any pre-trained molecular encoder**, capturing rich chemical

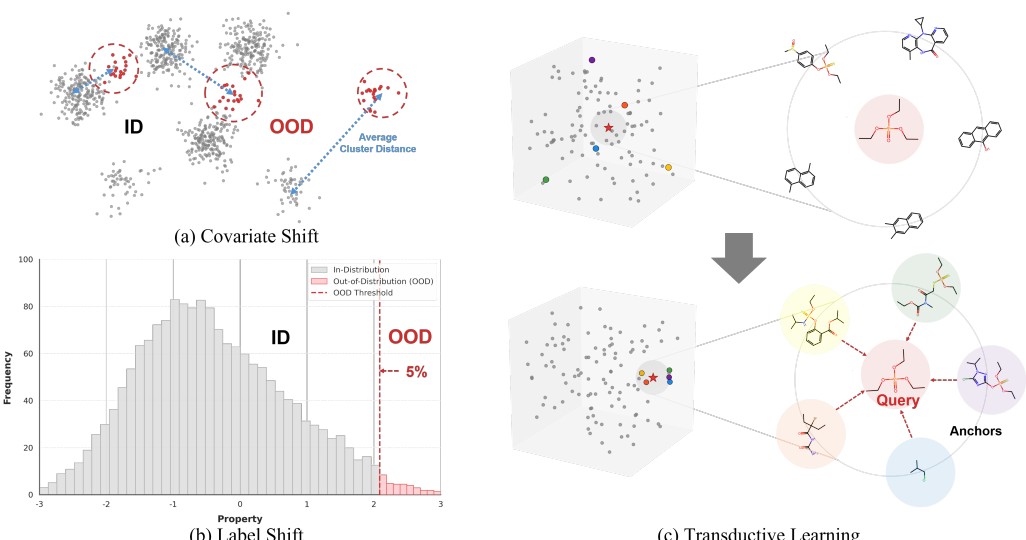

Figure 1: OOD challenges include: (a) covariate shift, where the model must generalize to molecules with new chemical structures, and (b) label shift, where the model must extrapolate to predict properties outside its training range. (c) MALT's transductive mechanism uses a memory bank of training examples to find multiple relevant anchors for given query molecule and particular task. By fusing the query's features with those of its closest anchors in a learned latent space, MALT produces more robust and accurate OOD predictions

knowledge. Instead of a single analogy, our method identifies and reasons with multiple relevant anchor molecules from the training data, dynamically fusing information from the query and its anchors to enable a more robust and nuanced prediction.

Our contributions are as follows:

- We introduce a novel, model-agnostic transductive framework that enhances existing molecular encoders by performing analogical reasoning in learned latent spaces, overcoming the fixed-descriptor limitations of prior work to enable richer chemical representations.
- We propose a multi-anchor latent reasoning mechanism that synthesizes information from multiple training analogs. This approach overcomes the brittleness of single-anchor methods, yielding more robust predictions by aggregating diverse chemical context.
- Through rigorous evaluation on practical OOD benchmarks, we demonstrate our framework systematically addresses both covariate and label shifts, improving OOD generalization. Ablation studies further validate the contributions of our multi-anchor design and training strategies, confirming their role in achieving robust performance that also matches or surpasses baseline results on ID tasks.

## 2 RELATED WORKS

### 2.1 THE CHALLENGE OF OUT-OF-DISTRIBUTION GENERALIZATION

ML models excel at interpolation within their training domain but struggle to extrapolate reliably. The ability to extrapolate is critical for discovering molecules with exceptional properties and is invaluable for adapting to proprietary chemical spaces (13; 14). Developing models that overcome these OOD limitations is therefore essential for trustworthy ML-driven discovery. Models often fail when encountering data that differs significantly from their training distribution because they are typically developed under the assumption that data is independently and identically distributed (IID) (6). This assumption is frequently violated in real-world applications, where models must

contend with novel chemical structures, unseen experimental conditions, or new property ranges. This distribution shift, depicted in Figure 1(a), can cause severe performance degradation and yield unreliable, overconfident predictions, limiting the practical utility of these models (3; 4). While general OOD techniques exist for graph-structured data (15), the unique complexities of molecular data necessitate domain-specific solutions.

In molecular modeling, OOD shifts can stem from multiple sources, such as variations in not only in core molecular scaffolds but to molecular size, biological targets, or experimental protocols (2; 6; 16) . These shifts manifest in two primary forms (1):

1. Covariate Shift: The distribution of molecular features, $P(X)$, changes. This typically occurs when the model must predict properties for novel chemical structures not represented in the training set (Figure 1(a), Top).

2. Label Shift: The marginal distribution of the target property, $P(Y)$, changes. This is relevant when extrapolating to property values that are rare or entirely absent in the training data (Figure 1(a), Bottom). In this scenario, the underlying relationship $P(Y|X)$ is assumed to be stable, but the model must make accurate predictions for Y values in sparsely populated regions of the target space.

## 2.2 STRATEGIES FOR OOD MOLECULAR PROPERTY PREDICTION(MPP)

Much of the work on out-of-distribution (OOD) molecular modeling has aimed to improve the robustness of inductive models. One common strategy is learning invariant representations (6; 17), which seeks to identify stable, predictive structural features across different data environments. However, capturing true invariance is difficult and often requires specific environmental labels that aren't always available. Another approach, uncertainty quantification (UQ) (3), uses prediction confidence to detect potential OOD failures. While useful for flagging unreliable predictions, UQ doesn't inherently improve the model's accuracy on OOD samples without further adaptation (18), and some methods can be resource-intensive and complex (19). A third strategy involves leveraging unlabeled data through methods like self-supervised or semi-supervised learning (7). For instance, domain-adaptive pre-training (DAPT) can enhance encoder representations by continuing pre-training on data similar to the target task. Other techniques use pseudo-labeling to integrate unlabeled data directly into training, though this risks propagating errors from inaccurate labels (20). While these methods improve encoders through more training, our work introduces a complementary, transductive approach that enhances them via analogical reasoning at inference time.

## 2.3 TRANSDUCTIVE STRATEGIES FOR OOD EXTRAPOLATION

Transductive learning aids OOD extrapolation by incorporating test queries $X_{test}$ alongside training data $\{(X_{train}, Y_{train})\}$. It often employs analogical extrapolative reasoning, comparing a query $x$ to training examples $x'$, improving robustness for OOD inputs or target values beyond the training range. One transductive approach is Test-Time Adaptation (TTA) (12; 21; 22), which adapts model parameters using unlabeled test batches to handle domain shifts. TTA focuses on model adaptation rather than direct prediction via training analogies and can be computationally intensive at inference.

Bilinear Transduction (9; 11) is a transductive method that formalizes analogical reasoning for out-of-distribution (OOD) prediction. This approach enables extrapolation by learning how properties change as functions of compositional differences, rather than learning direct mappings from materials to properties. Instead of learning a direct mapping $h : \mathcal{X} \to \mathcal{Y}$, it learns to predict a target value $y_i$ from an anchor point $x_j$ and their difference vector $\Delta x = x_i - x_j$ using a bilinear predictor of the form:

$$h_\theta(\Delta x, x) = f_\theta(\Delta x) \odot g_\theta(x) \tag{1}$$

where $f_\theta$ and $g_\theta$ are neural networks and $\odot$ represents element-wise multiplication.

The model is trained using only pairs where the anchor has lower property values than the target $(y_j < y_i)$, ensuring the model learns to extrapolate from lower to higher property values. The training objective minimizes:

$$\mathcal{L}(\theta) = \sum_{i=1}^{n} \sum_{j:y_j < y_i} \ell(h_\theta(x_i - x_j, x_j), y_i) \tag{2}$$

where the difference vectors form the constrained set $D_{\Delta X}^{tr} = \{x_i - x_j \mid x_i, x_j \in D_X^{tr}, y_j < y_i\}$.

For a test query $x_{te}$, an anchor $x_{an}$ is selected from the training set by minimizing the distance to the nearest training difference:

$$x_{an} = \arg \min_{x_{an} \in D_X^{tr}} \{\|x_{te} - x_{an} - \Delta x_{tr}\|_2 \mid \Delta x_{tr} \in D_{\Delta X}^{tr}\} \qquad (3)$$

The final prediction is then calculated as $y_{te} = h_\theta(x_{te} - x_{an}, x_{an})$.

Key limitations of this approach include: (1) reliance on hand-crafted descriptors rather than learned representations, (2) no uncertainty quantification in anchor selection, leading to potential brittleness from using a single anchor point, and (3) often reduced in-distribution accuracy compared to standard regression approaches.

Our proposed framework addresses these shortcomings by employing multi-anchor reasoning directly within learned latent representations derived from powerful molecular encoders, aiming for more robust and nuanced analogical predictions. We provide a detailed qualitative (Appendix A) and systematic chemical analysis (Appendix B) of anchors. Together, these sections highlight our model's ability to select informative and diverse anchors under distribution shifts—demonstrating a key strength of our approach.

## 3 METHODS

Our proposed framework augments standard inductive molecular property predictors with a transductive component operating in the latent space. The core idea is to leverage similarities between a query molecule embedding and those of multiple anchor molecules from the training set to improve prediction. The framework consists of three main components: an "arbitrary" molecular encoder (responsible for memory bank creation), a latent-space multi-anchor selection component (the transduction module), and a multi-anchor prediction head. The detailed pseudocode for our training and inference procedures can be found in Appendix C. An overview of the model architecture is presented in Appendix D. The choice to use multiple anchors is theoretically motivated, as this fusion mechanism can be shown to achieve a tighter test error bound compared to single-anchor methods, thereby improving OOD generalization (see Appendix K for the full derivation). A detailed analysis of the framework's computational overhead is provided in Appendix M.

### 3.1 MEMORY BANK

The foundation for the transductive component is a memory bank $Z_{train}$ containing fixed-dimensional latent embeddings $z_i \in \mathbb{R}^d$ for all molecules $(m_i, y_i)$ in the training set $\mathcal{D}_{train}$. This allows the framework to explicitly leverage similarities within the training data during prediction by serving as the source for anchor point selection.

Embeddings $z = \mathcal{E}(M)$ are generated using a modular molecular encoder $\mathcal{E}$. The framework design allows substituting **any arbitrary architecture**. In our experiments, we utilized several molecular encoders, including pretrained Graph Isomorphism Network (GIN) models (23), pretrained sequence-based models operating on SMILES (24), and a widely used Message Passing Neural Network (MPNN) model (25). The encoder can be initialized using various strategies, such as loading pretrained weights, using weights previously fine-tuned on $\mathcal{D}_{train}$, or starting from random initialization.

Crucially, unless explicitly frozen, the encoder $\mathcal{E}$ is trained end-to-end with the transduction module and prediction head components, allowing its parameters $\theta_{\mathcal{E}}$ to be updated during the main training loop (Refer to Table 8 for more information regarding training strategies). Consequently, the memory bank embeddings $Z_{\text{train}}$ must remain consistent with the evolving encoder. Thus, $Z_{\text{train}}$ is periodically regenerated (e.g., every $N$ epochs) by re-applying the updated encoder $\mathcal{E}$ to all training molecules $M_i \in \mathcal{D}_{\text{train}}$ (represented as SMILES (26) or graphs, depending on the encoder). Despite updating the memory bank every epoch, we did not observe a prohibitive increase in overall training time or computational requirements (refer to Appendix M for detailed computational overload analysis).

The memory bank used by the transduction module at any given point is thus formally defined based on the current encoder state:

$$Z_{train} = \{z_i = \mathcal{E}(m_i; \theta_{\mathcal{E}}) \mid (m_i, y_i) \in \mathcal{D}_{train}\} \tag{4}$$

### 3.2 Transduction Module: Latent-Space Multi-Anchor Selection

Given a query molecule $m_{query}$ and its latent embedding $z_{query} = \mathcal{E}(m_{query})$, the transduction module $\mathcal{T}$ selects $k$ anchor embeddings $Z_{\text{anchors}} = \{z_{a_1}, \ldots, z_{a_k}\}$ from the memory bank $Z_{train}$. This selection is based on similarity or distance metrics (e.g., cosine similarity and Euclidean distance) calculated between the query $z_{query}$ and the embeddings $z_i$ within $Z_{train}$. The transduction module outputs the selected anchor embeddings $Z_{\text{anchors}}$ and potentially their corresponding similarity/distance scores $W_{\text{anchors}}$ relative to the query. In our experiments we chose Top-K with Euclidean distance as our default choice. A detailed comparison and explanation of anchor selection methods experimented is presented in Appendix E, which also outlines the evaluation process and provides the rationale for our default choice.

### 3.3 Multi-Anchor Prediction Head

The prediction head $\mathcal{P}$ integrates information from the query embedding $z_{query}$ and the selected anchor embeddings $Z_{\text{anchors}}$ to produce the final property prediction $\hat{y}_{query}$. We employ a multi-head cross-attention mechanism where $z_{query}$ attends to the anchor embeddings $Z_{\text{anchors}}$ (serving as keys and values) to derive an attended anchor representation $z_{\text{attn}}$.

$$z_{\text{attn}} = \text{MultiHeadAttention}(Q = z_{query}, K = Z_{\text{anchors}}, V = Z_{\text{anchors}}) \tag{5}$$

This attended representation $z_{\text{attn}}$ is then combined with the original query embedding $z_{query}$. Optionally, the original anchor similarity/distance scores $W_{\text{anchors}}$ can also be incorporated at this stage to provide the final layers with explicit information about the relevance of each selected anchor. The resulting fused representation (containing information from $z_{query}$, $z_{attn}$, and potentially $W_{\text{anchors}}$) is processed through subsequent layers (e.g., an MLP consisting of linear layers and activation functions) to produce the final prediction $\hat{y}_{query}$. The advantage of using multi-anchors and our prediction head is further analyzed in the ablation studies.

## 4 Performance Evaluation

### 4.1 Experimental Settings

**Datasets** To rigorously evaluate our framework, we selected datasets from three distinct and complementary benchmarks designed to test performance under various distribution shifts, ensuring a thorough assessment across diverse chemical properties and demanding OOD scenarios.

- MoleculeNet (27): Widely used benchmark for its broad range of properties, including quantum mechanics, physical chemistry, biophysics, and physiology of molecules . We selected standard regression and classification tasks, aligning with methodologies from prior work (9).
- DrugOOD (2): To focus on targeted OOD challenges in drug discovery, we employed this systematic curator for drug target binding affinity prediction . We specifically used its curated $IC_{50}$ and $EC_{50}$ datasets with a scaffold splitting strategy for classification, following the setup in (17).
- Activity Cliffs (28): To assess performance on a particularly difficult OOD challenge, we used the activity cliffs benchmark. This scenario tests a model on structurally similar compounds that exhibit large differences in potency. The OOD test set for this benchmark consists of molecule pairs with high structural similarity but at least a tenfold difference in potency, evaluated across 30 different macromolecular targets.

**Splits** To rigorously evaluate OOD generalization, we employ a variety of data splitting strategies that induce different types of distributional shifts. These include standard approaches for structural

novelty (covariate shift) and property value extrapolation (label shift), as well as more complex, chemically meaningful scenarios designed to mimic real-world drug discovery challenges.

For inducing standard **covariate shifts**, which test generalization to new chemical structures, we follow established methodologies. For MoleculeNet datasets, we adopt the approach from (29), which uses spectral clustering on molecular cyclic skeletons to identify and separate the most structurally dissimilar molecules into an OOD test set. As validated in Appendix F, this method effectively creates a structural divide. For the DrugOOD classification task (2), we also include experiments where covariate shift is defined by molecular size (2, DrugOOD$_{ori}$)).

To address the critique that standard scaffold splits may not fully capture the complexity of real-world challenges, we incorporated two additional, more practical OOD scenarios:

- Activity Cliffs: Following (28), the OOD test set is constructed from molecule pairs with high structural similarity but at least a tenfold difference in potency.
- Lo-Hi Benchmark: This split simulates two distinct stages of a drug discovery campaign: Hit Identification (HI) and Lead Optimization (LO). Following (30), this setup provides a more realistic assessment of a model's utility in a prospective drug discovery pipeline.

For evaluating **label shift** extrapolation, we adopt the straightforward strategy from (9). In this setup, the OOD test set consists of molecules possessing the highest target property values, specifically those falling within the top 5% of the dataset's target value range. An accompanying ID test set is created by randomly sampling from the remaining 95% of the data.

**Evaluation metrics**   Following the evaluation criteria adopted by baseline models, we use AUROC to evaluate performance on classification tasks, including DrugOOD(IC$_{50}$, EC$_{50}$) and MoleculeNet (BBBP, ClinTox, SIDER). For regression tasks, we report MAE for MoleculeNet (BACE, ESOL, FreeSolv, Lipophilicity) and Lo-Hi benchmarks. For the Activity Cliffs benchmark, we report RMSE following prior work.

**Baselines**   To evaluate the effectiveness of our proposed framework, we compare its performance against several relevant baselines.

We include standard inductive-learning based MPP models. These represent the performance achievable using the base encoders without the transductive augmentation and were fine-tuned and evaluated under the same experimental conditions as our proposed method:

- Chemprop (25): A widely used directed message-passing neural network architecture for MPP.
- Pretrained GIN (23): A Graph Isomorphism Network(GIN) model, pre-trained on both supervised graph-level property prediction and atom-level context prediction as self-supervised pre-training strategies, learning from a dataset of 2 million molecules sourced from ZINC15 (31).
- SMI-TED (24): A Transformer-based encoder-decoder pre-trained using self-supervised learning on a large, curated dataset from PubChem (32) containing 91 million SMILES strings.
- UniMol (33): SE(3)-equivariant Transformer pretrained on 209M molecular conformations and 3.2M protein pockets with 3D position recovery + masked-atom prediction, finetuned on each dataset for property prediction.
- iMoLD (17): A framework for learning invariant molecular representations in a latent discrete space. It employs a "first-encoding-then-separation" strategy with an encoding GNN and a residual vector quantization module, along with a task-agnostic self-supervised learning objective to enhance out-of-distribution generalization.

We also include a transductive framework as a baseline, which is evaluated using either standard chemical features or embeddings from the inductive-learning models mentioned above:

- Bilinear Transduction (BLT, (9)): Rooted from (11), learns analogies between differences in RDKit (34) descriptors and corresponding property changes. This mechanism enables

Table 1: Performance (MAE) for $\text{Test}_{\text{ID}}$ and $\text{Test}_{\text{OOD}}$ Across Covariate Shift (X-Split) and Label Shift (Y-Split) Regression Benchmarks. Lower is better. For each split type and method group, best result is bolded, second best is underlined. Green background indicates improvement over inductive counterpart. Orange background indicates improvement over baseline transductive learning.

| Method | Embedding | BACE | | ESOL | | FreeSolv | | Lipophilicity | |
|---|---|---|---|---|---|---|---|---|---|
| | | $\text{Test}_{\text{ID}}$ | $\text{Test}_{\text{OOD}}$ | $\text{Test}_{\text{ID}}$ | $\text{Test}_{\text{OOD}}$ | $\text{Test}_{\text{ID}}$ | $\text{Test}_{\text{OOD}}$ | $\text{Test}_{\text{ID}}$ | $\text{Test}_{\text{OOD}}$ |
| **Covariate Shift (X-Splits)** | | | | | | | | | |
| *Inductive* | | | | | | | | | |
| | Chemprop | $0.5001_{\pm 0.0113}$ | $0.8848_{\pm 0.0403}$ | $0.5117_{\pm 0.0214}$ | $0.5117_{\pm 0.0214}$ | $0.1992_{\pm 0.0079}$ | $0.3369_{\pm 0.0269}$ | $0.3949_{\pm 0.0098}$ | $0.5097_{\pm 0.0191}$ |
| | GIN | $0.4727_{\pm 0.0094}$ | $0.7730_{\pm 0.0314}$ | $0.5415_{\pm 0.0178}$ | $0.5415_{\pm 0.0178}$ | $0.2928_{\pm 0.0122}$ | $0.3957_{\pm 0.0347}$ | $0.4461_{\pm 0.0068}$ | $0.5741_{\pm 0.0093}$ |
| | SMI-TED | $0.3615_{\pm 0.0189}$ | $0.6939_{\pm 0.0265}$ | $0.4508_{\pm 0.0178}$ | $0.4508_{\pm 0.0178}$ | $0.2401_{\pm 0.0110}$ | $0.3470_{\pm 0.0227}$ | $0.4275_{\pm 0.0136}$ | $0.5556_{\pm 0.0180}$ |
| | iMoLD | $0.4283_{\pm 0.0161}$ | $0.8658_{\pm 0.0743}$ | $0.2713_{\pm 0.0257}$ | $0.5941_{\pm 0.0497}$ | $0.2450_{\pm 0.0209}$ | $0.4297_{\pm 0.1343}$ | $0.4526_{\pm 0.0288}$ | $0.5958_{\pm 0.0264}$ |
| | UniMol | $0.3760_{\pm 0.1131}$ | $0.6130_{\pm 0.0198}$ | $0.1740_{\pm 0.0311}$ | $0.3930_{\pm 0.0139}$ | $0.1740_{\pm 0.0169}$ | $0.2750_{\pm 0.0237}$ | $0.3500_{\pm 0.0238}$ | $0.4370_{\pm 0.0031}$ |
| *Transductive (BLT)* | | | | | | | | | |
| | RDKit | $0.5649_{\pm 0.0455}$ | $0.6667_{\pm 0.0683}$ | $0.3850_{\pm 0.0297}$ | $\underline{0.3850}_{\pm 0.0297}$ | $0.1787_{\pm 0.0082}$ | $0.2850_{\pm 0.0452}$ | $0.5120_{\pm 0.0124}$ | $0.6252_{\pm 0.0259}$ |
| | GIN | $0.7585_{\pm 0.0328}$ | $0.9279_{\pm 0.0507}$ | $0.7383_{\pm 0.0296}$ | $0.7383_{\pm 0.0296}$ | $0.5244_{\pm 0.0353}$ | $0.4348_{\pm 0.0500}$ | $0.6908_{\pm 0.0248}$ | $0.9009_{\pm 0.0232}$ |
| | SMI-TED | $0.6578_{\pm 0.0537}$ | $0.7914_{\pm 0.1152}$ | $0.9769_{\pm 0.1049}$ | $0.9769_{\pm 0.1049}$ | $0.3762_{\pm 0.0433}$ | $0.3795_{\pm 0.0558}$ | $0.7080_{\pm 0.0106}$ | $0.7810_{\pm 0.0284}$ |
| *Transductive (Ours)* | | | | | | | | | |
| | MALT-RDKit | $0.3306_{\pm 0.0236}$ | $0.6079_{\pm 0.0530}$ | $0.2188_{\pm 0.0120}$ | $\mathbf{0.3658}_{\pm 0.0120}$ | $\mathbf{0.1266}_{\pm 0.0081}$ | $\underline{0.2391}_{\pm 0.0163}$ | $0.3879_{\pm 0.0091}$ | $0.6138_{\pm 0.0091}$ |
| | MALT-Chemprop | $\mathbf{0.2847}_{\pm 0.0165}$ | $0.7783_{\pm 0.0553}$ | $0.2180_{\pm 0.0049}$ | $0.5072_{\pm 0.0120}$ | $\underline{0.1522}_{\pm 0.0043}$ | $0.2999_{\pm 0.0164}$ | $0.3474_{\pm 0.0135}$ | $0.4894_{\pm 0.0263}$ |
| | MALT-GIN | $\underline{0.3317}_{\pm 0.0083}$ | $\underline{0.6333}_{\pm 0.0347}$ | $\underline{0.2103}_{\pm 0.0113}$ | $0.5305_{\pm 0.0120}$ | $0.1919_{\pm 0.0126}$ | $0.3388_{\pm 0.0255}$ | $\underline{0.3370}_{\pm 0.0007}$ | $0.5369_{\pm 0.0007}$ |
| | MALT-SMI-TED | $0.3037_{\pm 0.0046}$ | $0.6716_{\pm 0.0569}$ | $\mathbf{0.2057}_{\pm 0.0052}$ | $0.4322_{\pm 0.0118}$ | $0.1584_{\pm 0.0247}$ | $0.2613_{\pm 0.0329}$ | $0.3608_{\pm 0.0120}$ | $0.5417_{\pm 0.0164}$ |
| | MALT-UniMol | $0.3430_{\pm 0.0179}$ | $\mathbf{0.4340}_{\pm 0.0120}$ | $0.1670_{\pm 0.0067}$ | $0.3920_{\pm 0.0092}$ | $0.1660_{\pm 0.0082}$ | $\mathbf{0.2710}_{\pm 0.0013}$ | $\mathbf{0.3050}_{\pm 0.0156}$ | $\mathbf{0.4300}_{\pm 0.0046}$ |
| **Label Shift (Y-Splits)** | | | | | | | | | |
| *Inductive* | | | | | | | | | |
| | Chemprop | $0.4509_{\pm 0.0092}$ | $1.1331_{\pm 0.0410}$ | $0.1955_{\pm 0.0057}$ | $\underline{0.4506}_{\pm 0.0319}$ | $0.1967_{\pm 0.0083}$ | $0.3931_{\pm 0.0432}$ | $0.3560_{\pm 0.0080}$ | $0.6801_{\pm 0.0272}$ |
| | GIN | $0.4976_{\pm 0.0090}$ | $0.7343_{\pm 0.0235}$ | $0.2356_{\pm 0.0076}$ | $0.5293_{\pm 0.0086}$ | $0.2486_{\pm 0.0246}$ | $0.5544_{\pm 0.0229}$ | $0.3886_{\pm 0.0037}$ | $0.7241_{\pm 0.0135}$ |
| | SMI-TED | $0.3676_{\pm 0.0113}$ | $0.8741_{\pm 0.0660}$ | $0.2166_{\pm 0.0079}$ | $0.4607_{\pm 0.0272}$ | $0.3419_{\pm 0.0275}$ | $0.3954_{\pm 0.0620}$ | $0.3555_{\pm 0.0102}$ | $0.6499_{\pm 0.0578}$ |
| | iMoLD | $0.8107_{\pm 0.0019}$ | $1.5493_{\pm 0.0182}$ | $0.7219_{\pm 0.0016}$ | $1.7868_{\pm 0.0112}$ | $0.6953_{\pm 0.0047}$ | $1.6511_{\pm 0.0127}$ | $0.7758_{\pm 0.0005}$ | $1.5294_{\pm 0.0095}$ |
| | UniMol | $0.3680_{\pm 0.0312}$ | $1.0040_{\pm 0.0101}$ | $\underline{0.1750}_{\pm 0.0291}$ | $0.5470_{\pm 0.0032}$ | $0.1660_{\pm 0.0019}$ | $\underline{0.1680}_{\pm 0.0131}$ | $0.3260_{\pm 0.0338}$ | $0.6840_{\pm 0.0039}$ |
| *Transductive (BLT)* | | | | | | | | | |
| | RDKit | $0.5864_{\pm 0.1217}$ | $1.0728_{\pm 0.1962}$ | $0.2422_{\pm 0.0103}$ | $0.5132_{\pm 0.0295}$ | $0.3534_{\pm 0.0299}$ | $0.5124_{\pm 0.0327}$ | $0.4320_{\pm 0.0253}$ | $0.8367_{\pm 0.0332}$ |
| | GIN | $0.7169_{\pm 0.0322}$ | $1.2719_{\pm 0.0345}$ | $0.4860_{\pm 0.0328}$ | $0.9057_{\pm 0.0197}$ | $0.4588_{\pm 0.0206}$ | $0.7246_{\pm 0.0402}$ | $0.6679_{\pm 0.0166}$ | $1.1019_{\pm 0.0393}$ |
| | SMI-TED | $0.6006_{\pm 0.0996}$ | $1.2039_{\pm 0.1153}$ | $0.4536_{\pm 0.0516}$ | $0.9019_{\pm 0.0771}$ | $0.4683_{\pm 0.0864}$ | $0.9240_{\pm 0.0595}$ | $0.6360_{\pm 0.0649}$ | $1.1211_{\pm 0.0672}$ |
| *Transductive (Ours)* | | | | | | | | | |
| | MALT-RDKit | $0.3819_{\pm 0.0254}$ | $0.7833_{\pm 0.0285}$ | $0.1862_{\pm 0.0077}$ | $0.4906_{\pm 0.0172}$ | $\underline{0.1492}_{\pm 0.0080}$ | $0.3305_{\pm 0.0503}$ | $0.3246_{\pm 0.0103}$ | $0.7430_{\pm 0.0103}$ |
| | MALT-Chemprop | $\underline{0.3461}_{\pm 0.0097}$ | $0.7705_{\pm 0.0287}$ | $\mathbf{0.1734}_{\pm 0.0070}$ | $0.4994_{\pm 0.0114}$ | $\mathbf{0.1469}_{\pm 0.0179}$ | $0.2753_{\pm 0.0530}$ | $\underline{0.3185}_{\pm 0.0104}$ | $0.6444_{\pm 0.0151}$ |
| | MALT-GIN | $0.3861_{\pm 0.0097}$ | $\mathbf{0.7340}_{\pm 0.0121}$ | $0.1845_{\pm 0.0069}$ | $0.4989_{\pm 0.0267}$ | $0.1585_{\pm 0.0165}$ | $0.2637_{\pm 0.0220}$ | $0.3195_{\pm 0.0108}$ | $\mathbf{0.4690}_{\pm 0.0108}$ |
| | MALT-SMI-TED | $0.3546_{\pm 0.0164}$ | $0.8326_{\pm 0.0141}$ | $0.1852_{\pm 0.0096}$ | $0.5390_{\pm 0.0199}$ | $0.1716_{\pm 0.0061}$ | $0.2856_{\pm 0.0427}$ | $0.3300_{\pm 0.0097}$ | $0.6609_{\pm 0.0331}$ |
| | MALT-UniMol | $\mathbf{0.3440}_{\pm 0.0213}$ | $0.8220_{\pm 0.0146}$ | $0.2030_{\pm 0.0121}$ | $\mathbf{0.4190}_{\pm 0.0047}$ | $0.1610_{\pm 0.0031}$ | $\mathbf{0.1200}_{\pm 0.0038}$ | $\mathbf{0.3080}_{\pm 0.0011}$ | $\underline{0.5850}_{\pm 0.0027}$ |

Table 2: Performance (AUROC) for $\text{Test}_{\text{OOD}}$ Covariate Shift (X-Split) Classification. For DrugOOD, we follow the split done in (2) ($\text{DrugOOD}_{\text{ori}}$). We also include the results following the split introduced in (29)($\text{DrugOOD}_{\text{ours}}$, $\text{MoleculeNet}_{\text{ours}}$ ). Higher values are better. Best results bolded, second best underlined. Green background shows improvement over inductive models; Orange background shows RDKit (Ours) improvement over baseline transductive learning.

| Method | Embedding | $\text{DrugOOD}_{\text{ori}}$ | | $\text{DrugOOD}_{\text{ours}}$ | | $\text{MoleculeNet}_{\text{ours}}$ | | |
|---|---|---|---|---|---|---|---|---|
| | | EC50 | IC50 | EC50 | IC50 | BBBP | ClinTox | SIDER |
| *Inductive* | | | | | | | | |
| | Chemprop | $0.6423 \pm 0.0041$ | $0.6577 \pm 0.0011$ | $0.7614 \pm 0.0325$ | $0.8132 \pm 0.0139$ | $0.7719 \pm 0.1097$ | $0.9483 \pm 0.0114$ | $0.6150 \pm 0.0514$ |
| | GIN | $0.6632 \pm 0.0076$ | $\mathbf{0.6866 \pm 0.0013}$ | $0.7696 \pm 0.0096$ | $0.8424 \pm 0.0032$ | $0.7610 \pm 0.0206$ | $0.8102 \pm 0.0190$ | $0.4747 \pm 0.0198$ |
| | SMI-TED | $0.5912 \pm 0.0376$ | $0.6367 \pm 0.0252$ | $0.7094 \pm 0.0325$ | $0.6367 \pm 0.0252$ | $0.5948 \pm 0.1802$ | $0.9020 \pm 0.1146$ | $0.5513 \pm 0.0346$ |
| | iMoLD | $0.6884 \pm 0.0058$ | $0.6779 \pm 0.0088$ | $0.7821 \pm 0.0188$ | $0.7873 \pm 0.0131$ | $0.8247 \pm 0.0608$ | $0.8973 \pm 0.0284$ | $0.6550 \pm 0.0307$ |
| *Transductive (BLT)* | | | | | | | | |
| | RDKit | $0.5864 \pm 0.0044$ | $0.6239 \pm 0.0097$ | $0.5638 \pm 0.0091$ | $0.7457 \pm 0.0057$ | $0.6807 \pm 0.0372$ | $0.7616 \pm 0.1296$ | $0.5768 \pm 0.0658$ |
| | GIN | $0.5962 \pm 0.0079$ | $0.5588 \pm 0.0023$ | $0.7076 \pm 0.0084$ | $0.6626 \pm 0.0052$ | $0.5889 \pm 0.0600$ | $0.8102 \pm 0.0352$ | $0.6232 \pm 0.0619$ |
| | SMI-TED | $0.5895 \pm 0.0106$ | $0.6446 \pm 0.0060$ | $0.8030 \pm 0.0112$ | $0.7726 \pm 0.0044$ | $0.7686 \pm 0.0727$ | $0.8422 \pm 0.0504$ | $0.5542 \pm 0.0679$ |
| *Transductive (Ours)* | | | | | | | | |
| | MALT-RDKit | $0.6658 \pm 0.0141$ | $0.6659 \pm 0.0131$ | $0.7532 \pm 0.0236$ | $0.8055 \pm 0.0088$ | $0.8095 \pm 0.0240$ | $0.9395 \pm 0.0206$ | $\mathbf{0.8095 \pm 0.0240}$ |
| | MALT-Chemprop | $0.6485 \pm 0.0072$ | $\underline{0.6759 \pm 0.0057}$ | $0.7953 \pm 0.0187$ | $0.8330 \pm 0.0099$ | $0.8671 \pm 0.0108$ | $\mathbf{0.9517 \pm 0.0085}$ | $0.6727 \pm 0.0344$ |
| | MALT-GIN | $\mathbf{0.6959 \pm 0.0110}$ | $0.6632 \pm 0.0014$ | $\mathbf{0.8138 \pm 0.0147}$ | $\mathbf{0.8499 \pm 0.0106}$ | $0.8039 \pm 0.0193$ | $0.8122 \pm 0.0107$ | $0.5970 \pm 0.0351$ |
| | MALT-SMI-TED | $\underline{0.6826 \pm 0.0126}$ | $0.6694 \pm 0.0092$ | $0.7899 \pm 0.0161$ | $0.6524 \pm 0.0062$ | $\mathbf{0.8684 \pm 0.0279}$ | $0.9510 \pm 0.0071$ | $0.6057 \pm 0.0273$ |

extrapolation beyond the range of the training data. In our comparative analysis, we employ BLT not only with standard RDKit descriptors but also with embeddings obtained from two of the previously listed inductive models: Pretrained GIN and SMI-TED .

## 4.2 PERFORMANCE COMPARISON

Our framework demonstrates a consistent and significant improvement in OOD generalization across a wide array of regression and classification benchmarks. By operating in a learned latent space and

leveraging multiple anchors, MALT not only enhances the performance of strong inductive base models but also substantially outperforms existing transductive methods(Table 1, Table 2). To visually complement results for regression tasks, we provide parity plots in Appendix I that compare predicted versus true values and embedding space transformations of anchors in Appendix J. As shown in the parity plots, for a majority of the tasks, MALT increases performance and embedding space transformations show that important anchors cluster towards each other after transductive learning.

**Enhancing Inductive Models on Standard Benchmarks**  As shown in Table 1, augmenting standard inductive encoders (Chemprop, GIN, SMI-TED) with our transductive module consistently improves performance on both covariate (X-Splits) and label (Y-Splits) shift regression tasks. The improvements, highlighted in green, are evident across nearly all datasets for both ID and OOD test sets. This pattern holds for classification tasks under covariate shift (Table 2), where MALT boosts the AUROC of the base models. This confirms that our modular, transductive reasoning component effectively enhances the predictive power of various molecular encoders. In many cases, our augmented models achieve the best overall performance (indicated in **bold**).

**Outperforming Transductive Baselines**  MALT also demonstrates a clear advantage over the BLT baseline from (9). When using identical RDKit descriptors, our model (*Ours (RDKit)*) achieves a consistently lower MAE than *BLT (RDKit)* across all regression settings (highlighted in orange in Table 1) and a higher AUROC in all classification tasks (Table 2).

Crucially, while BLT struggles to effectively utilize the rich representations from pretrained GIN and SMI-TED embeddings—often performing worse than its own RDKit variant—our framework excels. MALT successfully integrates these advanced embeddings, leading to robust OOD performance and demonstrating a unique capability to adapt learned latent representations for transductive reasoning. A detailed analysis of this is shown in Appendix A and Appendix B.

**Realistic Drug Discovery Scenarios**  To validate our framework's practical utility beyond standard academic benchmarks, we evaluated it on more complex and chemically meaningful OOD scenarios. Results on these experiments additionally confirm MALT's robustness and effectiveness in settings that closely mimic real-world challenges.

- **Activity Cliffs:** We tested MALT on a challenging activity cliffs benchmark, where minor structural changes lead to large potency differences (28). As detailed in Appendix L, MALT-enhanced models achieved a top-2 rank far more frequently than their base counterparts across 30 pharmacological endpoints. This resulted in substantial median RMSE reductions of up to 12.7% for OOD data, showcasing MALT's ability to navigate difficult regions of the chemical space.

- **Lo-Hi Benchmark:** Our framework was further evaluated on the Lo-Hi benchmark (30), which simulates the Hit Identification (HI) and Lead Optimization (LO) stages of a drug discovery campaign. MALT-Chemprop consistently outperformed hyperparameter-tuned Chemprop baseline across most splits. Notably, MALT achieved performance gains of 31.58% on the FreeSolv LO split and over 20% on several scaffold-based splits (see Appendix L), validating its effectiveness in a realistic discovery pipeline.

### 4.3 Ablation Studies

To validate the key architectural and methodological choices of our framework, we conducted a series of ablation studies. These experiments systematically investigate the impact of the encoder training strategy, the necessity of the multi-anchor selection mechanism, the model's robustness to noisy information, and its advantages over simpler non-learning baselines. All corresponding result tables can be found in Appendix H.

**Importance of Jointly Training the Encoder and Transduction Module**  We first investigated the optimal training strategy for the molecular encoder $\mathcal{E}$ within our transductive framework. As shown in Table 8, we compared strategies where the encoder was either pre-finetuned on the task and/or adapted (i.e., its weights were updated) during the main transductive training phase. The results unequivocally show that the best performance is achieved with the 'Finetune O, Adapt O' strategy,

where a task-finetuned encoder is jointly trained with the transduction module. This confirms that allowing the encoder to adapt creates a more effective latent space that is optimized not just for the task, but for the analogical reasoning required by the transduction module.

**Multi-Anchor Selection Strategy Validation**  Our framework's core hypothesis is that using multiple, high-quality anchors is superior to single-anchor or arbitrary-anchor methods. The results in Table 9 strongly support this. The findings show that multiple anchors consistently outperform a single anchor, as performance improves with the number of anchors ($k$) increasing from 1 to 10. For instance, on the FreeSolv Y-Split for $Test_{OOD}$, using a single anchor ($k = 1$) results in an MAE of 0.3642, whereas our default strategy with 10 anchors achieves a significantly lower MAE of 0.2637. Furthermore, the relevance of the selected anchors is paramount, demonstrating that top-ranked anchors are essential. A model using the top 10 anchors for the Lipo Y-Split ($Test_{OOD}$) achieves an MAE of 0.4690. In contrast, a model using the same number of lower-ranked anchors ($11^{th}$ to $20^{th}$) performs much worse, with an MAE of 0.4879. This demonstrates that the learned embedding space is meaningful, correctly identifying the most informative analogies for prediction.

**Robustness to Noisy Anchors**  A potential failure mode for a multi-anchor system is sensitivity to noisy or irrelevant anchors. We tested MALT's resilience by deliberately replacing top-ranked anchors with the lowest-ranked ("noisiest") ones from the training set. As shown in Table 10, the framework demonstrates graceful degradation rather than catastrophic failure. For example, on the Lipo X-Split ($Test_{OOD}$), replacing 5 out of 10 anchors with the worst possible choices only increases MAE from 0.4736 to 0.4748. This resilience indicates that the model's attention mechanism successfully learns to discount the influence of irrelevant anchors, a crucial feature for robust real-world performance.

**Disentangling Representation and Reasoning from Simple Retrieval**  To prove our model learns more than a simple similarity search, we compared the full MALT-GIN against several k-nearest neighbor (k-NN) baselines that average property values. The results in Table 11 yield two insights. First, MALT learns a appropriate task-specific representation. A k-NN model using embeddings from our trained MALT-GIN outperforms k-NN using embeddings from the pretrained GIN or ECFP fingerprints. This confirms our end-to-end training produces a more effective latent space for the task. Second, the fusion mechanism adds value beyond retrieval. The full MALT-GIN model outperforms the k-NN baseline that uses its own powerful embeddings. This performance gap isolates the contribution of the attention-based fusion head, proving that the model's ability to intelligently weigh and integrate anchor information is critical to its success.

**Scalability on Large-Scale Datasets**  Finally, to confirm that our framework's advantages are not limited to smaller benchmarks, we evaluated it on the QM9 dataset (>133,000 molecules) for HOMO and LUMO prediction. The results in Table 12 show that MALT-enhanced models maintain their performance edge, outperforming their base inductive counterparts. This confirms that our approach scales effectively to large scientific datasets while preserving its robust performance benefits.

## 5 CONCLUSION

We introduced a multi-anchor transductive framework for molecular property prediction, designed to improve generalization in out-of-distribution settings. Operating in the latent space of pretrained encoders, our model-agnostic approach advances beyond inductive baselines and prior single-anchor transductive methods. By relating each target molecule to multiple training instances, the framework enables more robust and adaptive representation learning in novel chemical spaces and can be applied in a plug-and-play fashion to any arbitrary encoder.Comprehensive experiments and ablation studies confirm that MALT enhances both OOD generalization and in-distribution performance, consistently surpassing existing transductive baselines. Future work includes exploring more principled anchor selection, developing scalable search strategies to mitigate computational cost, and hybrid strategies with fallback mechanisms when transductive learning underperforms.

# 6    ETHICS STATEMENT

This work develops a multi-anchor transductive framework for molecular property prediction to accelerate drug discovery and materials science. While designed for beneficial applications, we acknowledge the dual-use potential—techniques enabling therapeutic compound discovery could theoretically be misused to design harmful substances. We emphasize the importance of responsible development and deployment of such predictive models.

Our research follows established ethical guidelines for computational chemistry and machine learning. All datasets are publicly available and properly cited, with no proprietary data or human subjects involved. We encourage practitioners to implement our framework within appropriate institutional oversight and regulatory frameworks, especially for sensitive applications like pharmaceutical development or chemical synthesis.

# 7    LLM USAGE

We used a large language model (LLM) as a general-purpose assistant for writing—suggesting phrasing, improving grammar and clarity, and helping with organization and citation formatting. The LLM also provided lightweight coding help (e.g., debugging minor errors and refactoring scripts); all ideas, analyses, and final text/code were created and verified by the authors.

# 8    REPRODUCIBILITY STATEMENT

Upon acceptance, the full code will be released publicly. We have released an anonymous github link and data link as well as our code in the supplementary materials. In the paper, we also provide comprehensive resources for reproduction. Complete hyperparameter configurations are in Table 7 and Appendix G. Our framework architecture is documented in Section 3, with additional details in Appendix D including algorithms for memory bank construction and inference. Appendix K includes a theoretical analysis and justification of our multi-anchor approach compared to bilinear transduction. Appendix E provides systematic evaluation of anchor selection methods, justifying our Top-K with Euclidean distance approach. All baseline implementations are detailed in Section 4.1. Data preprocessing and splitting methodologies are also covered in Section 4.1 and Appendix F, which validates our split method. Results include statistical reporting across multiple runs with different seeds, and with additional ablation studies in Section 4.3 and Appendix H. Computational overhead analysis is provided in Appendix M.

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

# A    QUALITATIVE ANCHOR ANALYSIS

To evaluate the effectiveness of multi-anchor reasoning within learned latent spaces, we compared the Top-5 anchors selected by our model and by BLT after training. As illustrated in Figure 2, we analyzed three representative molecules from the Test$_{OOD}$ dataset: (1) a randomly selected molecule (Random), (2) a molecule with the lowest average Maximum Common Substructure (MCS) similarity to training set scaffolds (Extreme Covariate Shift), and (3) a molecule with the highest target property value (Extreme Label Shift).

**(a) Random**

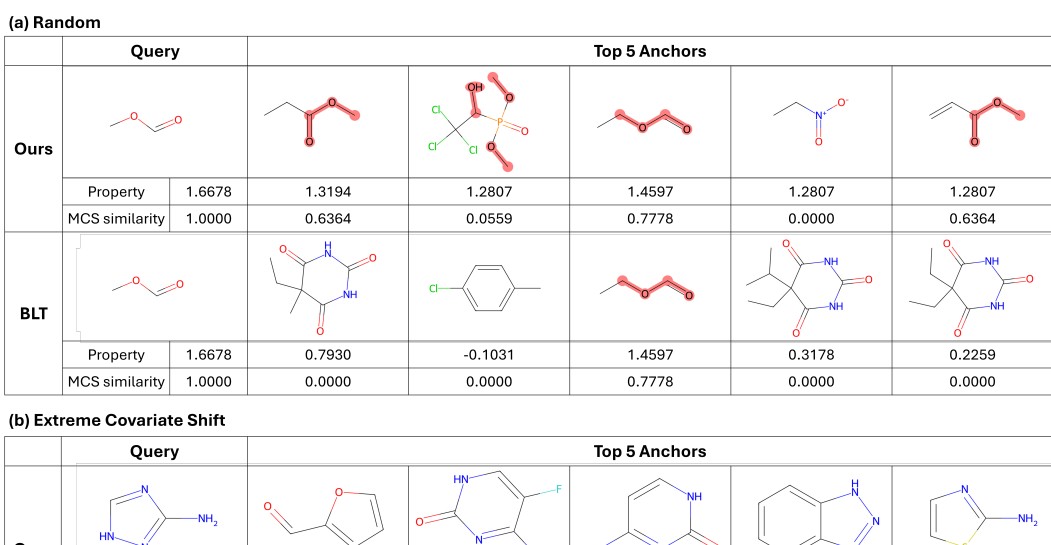

**(b) Extreme Covariate Shift**

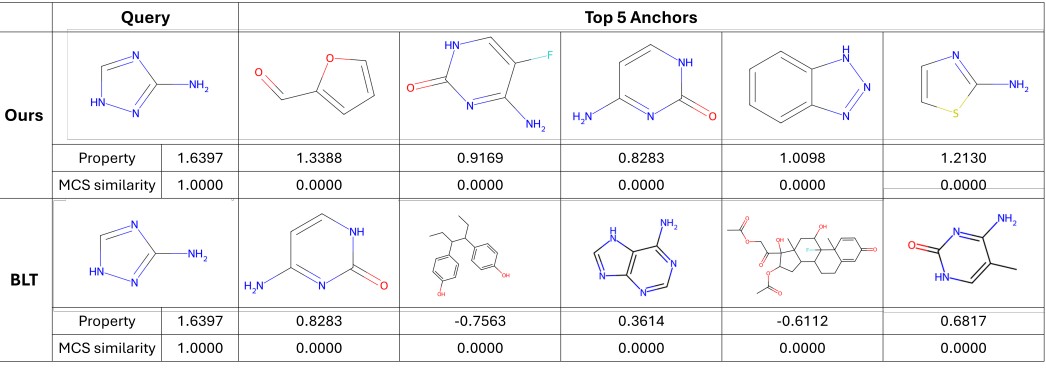

**(c) Extreme Label Shift**

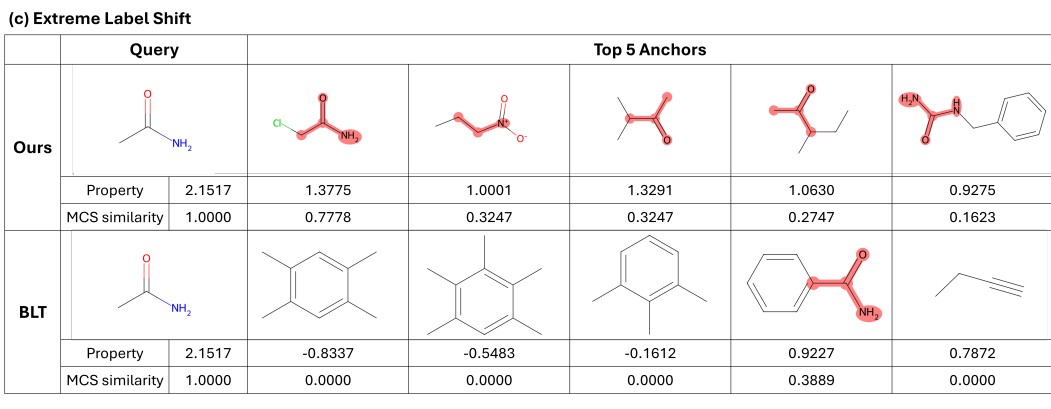

Figure 2: Comparison between Top 5 anchors retrieved by our model and BLT in ESOL dataset. Red highlights indicate the Maximum Common Substructure (MCS) between the query molecule and each corresponding anchor molecule. (a) Random: a randomly selected molecule from the Test$_{OOD}$ dataset. (b) Extreme Covariate Shift: a molecule from the Test$_{OOD}$ dataset with lowest average Maximum Common Substructure (MCS) similarity to Train set scaffolds. (c) Extreme Label Shift: a molecule from the Test$_{OOD}$ dataset with the highest target property value.

For each case, we assessed the alignment between the query and selected anchors in terms of both molecular property values and MCS similarity. Our method consistently selected anchors that were more functionally and structurally similar to the query compared to those selected by BLT. In contrast, BLT's selected anchors often exhibited greater divergence from the query in both dimensions. Notably, in the Extreme Covariate Shift case, our model demonstrated a clear advantage by capturing nuanced structural and chemical similarities that BLT—constrained by fixed input descriptors—failed to represent. This highlights the strength of learned latent representations in generalizing to structurally dissimilar molecules. In the Extreme Label Shift case, where no single anchor offers a perfect analogy, our model benefited from leveraging multiple complementary anchors. This multi-anchor strategy enabled the model to integrate diverse signals and make accurate predictions, whereas BLT's reliance on a single anchor limited its effectiveness. These results underscore the utility of multi-anchor reasoning in addressing the inherent limitations of single-anchor analogical inference.

## B    SYSTEMATIC CHEMICAL ANALYSIS OF ANCHOR SELECTION

To provide a quantitative understanding of the anchor selection mechanism, we perform a comprehensive chemical analysis comparing the training data, the out-of-distribution (OOD) query molecules, and the anchors selected by our model. This multifaceted analysis examines physicochemical properties, structural features, and quantitative similarity metrics, revealing the chemical principles that guide anchor selection.

### B.1    PHYSICOCHEMICAL PROPERTY ANALYSIS

We first compared the distributions of key physicochemical properties: molecular weight (MW), logarithm of the partition coefficient (LogP), and topological polar surface area (TPSA). The analysis, summarized in Table 3, covers both scaffold-based (X-split) and property-based (Y-split) OOD scenarios. The results consistently show that the model selects anchors with properties that are intermediate between the training distribution and the OOD queries. This suggests that the model does not merely select the most similar molecules but rather identifies anchors that chemically bridge the gap between the training and OOD domains.

Table 3: Physicochemical Property Comparison. For each dataset, results are presented for both the scaffold-based (X-split, left) and property-based (Y-split, right) OOD settings.

| Dataset | Property | X-Split | | | Y-Split | | |
|---|---|---|---|---|---|---|---|
| | | Train Mean (Median) | OOD (Query) Mean (Median) | Anchors Mean (Median) | Train Mean (Median) | OOD (Query) Mean (Median) | Anchors Mean (Median) |
| **BACE** | Mol. Weight | 481.7 (465.6) | 447.1 (452.0) | 476.3 (422.5) | 472.3 (457.5) | 617.4 (608.7) | 579.1 (579.8) |
| | LogP | 3.11 (3.12) | 3.62 (4.60) | 3.31 (3.52) | 3.12 (3.13) | 3.10 (3.30) | 3.36 (3.37) |
| | TPSA | 95.3 (91.2) | 97.5 (78.2) | 101.1 (85.6) | 93.5 (89.6) | 128.3 (111.9) | 113.1 (108.3) |
| **ESOL** | Mol. Weight | 196.6 (179.2) | 337.2 (307.3) | 275.9 (268.4) | 209.7 (192.0) | 108.5 (88.1) | 145.1 (108.1) |
| | LogP | 2.38 (2.30) | 3.58 (4.17) | 3.18 (3.40) | 2.58 (2.44) | 0.07 (0.39) | 0.61 (1.01) |
| | TPSA | 33.7 (26.0) | 56.4 (56.7) | 46.5 (41.6) | 42.1 (26.6) | 48.4 (28.7) | |
| **FreeSolv** | Mol. Weight | 134.8 (120.2) | 258.6 (241.3) | 123.9 (118.2) | 141.0 (122.1) | 102.9 (100.2) | 111.3 (99.0) |
| | LogP | 1.88 (1.74) | 3.83 (3.75) | 1.68 (1.56) | 1.91 (1.75) | 2.71 (2.76) | 2.29 (2.04) |
| | TPSA | 20.3 (17.1) | 18.7 (9.2) | 21.5 (20.2) | 21.0 (18.5) | 0.0 (0.0) | 0.2 (0.0) |
| **Lipo** | Mol. Weight | 387.2 (390.9) | 320.6 (305.8) | 404.8 (401.9) | 381.4 (386.5) | 417.1 (423.5) | 419.6 (427.4) |
| | LogP | 3.31 (3.31) | 2.71 (2.37) | 3.29 (3.22) | 3.22 (3.22) | 4.39 (4.25) | 3.90 (3.68) |
| | TPSA | 79.8 (80.7) | 65.6 (63.2) | 81.6 (83.1) | 79.4 (80.0) | 69.6 (69.2) | 75.8 (76.1) |

### B.2    STRUCTURAL AND FRAGMENT ANALYSIS

To investigate the structural basis of anchor selection, we analyzed the prevalence of molecular scaffolds and fragments using Murcko scaffolds, BRICS motifs, and RECAP fragments. The results, shown for the X-split in Tables 4 and 5, reveal that the selected anchors share significantly more relevant structural motifs with the OOD queries than a random sample from the training set would. This demonstrates that anchors are chosen for their fundamental structural relevance to the query molecule, providing a chemically sound basis for prediction.

Table 4: Top Scaffolds and Structural Diversity Metrics for the X-Split.

| Dataset | Data Type | Top 1 Scaffold (Count) | Top 2 Scaffold (Count) | Entropy | Diversity |
|---|---|---|---|---|---|
| BACE | Train | O=C1NC=NC1(c1ccccc1)c1ccccc1 (52) | O=S1(=O)CC(Cc2ccccc2)CC([NH2+]C c2ccccc2)C1 (50) | 8.327 | 0.445 |
| | OOD | c1ccc(-c2ccc(-c3ccccc3)n2Cc2ccc cn2)cc1 (14) | c1ccccc1 (7) | 4.173 | 0.527 |
| | Anchors | C1=NC2(CO1)c1ccccc1Oc1ccc(-c3cc ccc3)cc12 (105) | O=S1(=O)CC(Cc2ccccc2)CC([NH2+]C c2ccccc2)C1 (28) | 5.234 | 0.169 |
| ESOL | Train | No Scaffold (303) | c1ccccc1 (234) | 4.797 | 0.217 |
| | OOD | O=C(OCc1cccc(Oc2ccccc2)c1)C1CC1 (7) | O=c1oc2ccccc2cc1Cc1ccccc1 (2) | 5.280 | 0.833 |
| | Anchors | c1ccccc1 (77) | O=C1CC(=O)NC(=O)N1 (51) | 5.297 | 0.168 |
| FreeSolv | Train | No Scaffold (305) | c1ccccc1 (143) | 2.656 | 0.079 |
| | OOD | c1ccc(Cn2ccnc2)cc1 (1) | c1ccc(Cc2ccccc2)cc1 (1) | 3.907 | 1.000 |
| | Anchors | No Scaffold (92) | c1ccccc1 (31) | 1.901 | 0.080 |
| Lipo | Train | c1ccc(-c2ccccc2)cc1 (31) | O=C(Cc1ccccc1)NC1CCN(CCC(c2cccc c2)c2ccccc2)CC1 (28) | 10.539 | 0.587 |
| | OOD | c1ccccc1 (76) | c1ccnccc1 (6) | 5.000 | 0.462 |
| | Anchors | c1cnc(-c2ccc(C3CCCCC3)cc2)cn1 (53) | O=S(=O)(NCC(c1ccccc1)N1CCCCCC1) c1ccccc1 (47) | 8.373 | 0.291 |

Table 5: Analysis of Common BRICS Motifs and RECAP Fragments for the X-Split.

| Dataset | Data Type | BRICS Analysis | | | | RECAP Analysis | | | |
|---|---|---|---|---|---|---|---|---|---|
| | | Top 1 (Count) | Top 2 (Count) | Diversity | Entropy | Top 1 (Count) | Top 2 (Count) | Diversity | Entropy |
| BACE | Train | *N* (729) | *c1ccc(*)c1 (655) | - | - | *C(C)=O (205) | *S(C)(=O)=O (141) | 0.323 | 8.641 |
| | OOD | *C* (43) | *c1ccc(*)cc1 (29) | - | - | *c1ccc(*)n1* (25) | *Cc1cccc(N)n1 (20) | 0.382 | 5.403 |
| | Anchors | *N* (262) | *c1ccccc1 (247) | - | - | *c1ccccc1 (122) | *c1ccc(*)n1* (121) | 0.114 | 6.431 |
| ESOL | Train | *CC (131) | *O* (105) | 0.447 | 8.155 | *C(C)C (30) | *C(C)=O (29) | 0.474 | 7.614 |
| | OOD | *c1ccccc1 (16) | *O* (14) | 0.503 | 5.761 | *c1ccccc1 (10) | *CC1C(C(*)=O)C1(C)C (6) | 0.600 | 5.218 |
| | Anchors | *O* (97) | *CC (71) | 0.179 | 6.582 | *C(C)=O (39) | *C(C)C (28) | 0.280 | 6.492 |
| FreeSolv | Train | *CC (49) | *OC (45) | 0.578 | 7.991 | *C(C)=O (16) | *OC (15) | 0.460 | 6.249 |
| | OOD | *c1ccccc1 (4) | *C* (2) | 0.848 | 4.681 | *O (2) | *Cc1ccccc1 (1) | 0.917 | 3.418 |
| | Anchors | *O* (22) | *CC (15) | 0.398 | 5.943 | *C[C@H](C)O (6) | *CCCC (6) | 0.508 | 4.571 |
| Lipo | Train | *N* (2018) | *C(*)=O (1225) | 0.116 | 7.642 | *O (434) | *c1ccccc1 (335) | 0.286 | 9.959 |
| | OOD | *N* (62) | *C(*)=O (52) | 0.294 | 6.715 | *O (18) | *N1CCN(*)CC1 (16) | 0.556 | 7.508 |
| | Anchors | *N* (1093) | *c1ccccc1 (693) | 0.071 | 7.019 | *O (219) | *c1ccccc1 (206) | 0.175 | 8.778 |

Table 6: Tanimoto Similarity Between OOD Queries and Training Set Molecules' Morgan Fingerprints. For each dataset, results for the X-split and Y-split are shown, respectively.

| Similarity Type | Comparison | BACE | | ESOL | | FreeSolv | | Lipo | |
|---|---|---|---|---|---|---|---|---|---|
| | | X | Y | X | Y | X | Y | X | Y |
| Whole Molecule | Query vs. Anchor | **0.495** | **0.835** | **0.382** | **0.187** | 0.048 | **0.262** | **0.313** | **0.532** |
| | Query vs. Non-anchor | 0.328 | 0.438 | 0.094 | 0.046 | 0.054 | 0.067 | 0.220 | 0.291 |
| Scaffold | Query vs. Anchor | **0.351** | **0.808** | **0.314** | **0.567** | **0.075** | **0.762** | **0.180** | **0.516** |
| | Query vs. Non-anchor | 0.193 | 0.295 | 0.098 | 0.020 | 0.060 | 0.008 | 0.116 | 0.216 |

## B.3 QUANTITATIVE SIMILARITY COMPARISON

Finally, we quantified the similarity between OOD queries and their selected anchors using Tanimoto similarity with ECFP4 fingerprints. We compared this to the similarity between queries and all other non-anchor molecules in the training set. As shown in Table 6, the results demonstrate a clear and consistent pattern across all datasets and OOD splits.

These analyses reveal two global trends:

1. **Anchors are significantly more similar to queries than non-anchors.** This finding holds true across different datasets, OOD split types, and for both whole-molecule and scaffold-level similarity.

2. **The similarity gap is particularly large for scaffolds.** This highlights the model's ability to identify molecules with fundamentally similar core structures to serve as anchors, which is critical for making chemically sound and generalizable predictions.

# C ALGORITHM PSEUDOCODE

---

**Algorithm 1** Train

---

1: **Input:** Training data $\mathcal{D}_{\text{train}} = \{(m_i, y_i)\}$
2: **Components:** Encoder $\mathcal{E}$, Transduction Module $\mathcal{T}$, Multi-Anchor Prediction Head $\mathcal{P}$, Task Loss $\mathcal{L}_{\text{task}}$

3: **Initialize:** Parameters for $\mathcal{E}, \mathcal{T}, \mathcal{P}$; Optimizer; Scheduler
4: $Z_{\text{train}} \leftarrow \{\mathcal{E}(m_i) \mid (m_i, y_i) \in \mathcal{D}_{\text{train}}\}$      $\triangleright$ Initialize Memory Bank with initial $\mathcal{E}$

5: **for** epoch = 1 to Max Epochs **do**
6:      **if** epoch mod $N_{\text{update}} = 0$ **then**      $\triangleright$ Periodically update Memory Bank
7:          $Z_{\text{train}} \leftarrow \{\mathcal{E}(m_i) \mid (m_i, y_i) \in \mathcal{D}_{\text{train}}\}$      $\triangleright$ Use current $\mathcal{E}$
8:      **end if**
9:      **for** each batch $(M_{\text{batch}}, y_{\text{batch}})$ from $\mathcal{D}_{\text{train}}$ **do**
10:          $z_{\text{batch}} \leftarrow \mathcal{E}(M_{\text{batch}})$      $\triangleright$ Encode batch
11:          $Z_{\text{anchors}}, W_{\text{anchors}} \leftarrow \mathcal{T}(z_{\text{batch}}, Z_{\text{train}})$      $\triangleright$ Retrieve $k$ anchors and weights
12:          $\hat{y}_{\text{batch}} \leftarrow \mathcal{P}(z_{\text{batch}}, Z_{\text{anchors}}, W_{\text{anchors}})$      $\triangleright$ Multi-anchor prediction
13:          $\mathcal{L} \leftarrow \mathcal{L}_{\text{task}}(\hat{y}_{\text{batch}}, y_{\text{batch}})$      $\triangleright$ Compute batch loss
14:          Backpropagate $\mathcal{L}$ to update $\theta_{\mathcal{E}}, \theta_{\mathcal{P}}, \theta_{\mathcal{T}}$
15:      **end for**
16: **end for**
17: **return** Trained parameters $\theta_{\mathcal{E}}, \theta_{\mathcal{P}}$, Final Memory Bank $Z_{\text{train}}$

---

**Algorithm 2** Inference

---

1: **Input:** Final Memory Bank $Z_{\text{train}}$ , $m_{\text{query}}$
2: **Components:** Trained Molecular Encoder $\mathcal{E}$, Transduction Module $\mathcal{T}$, Trained Multi-Anchor Prediction Head $\mathcal{P}$

3: $z_{\text{query}} \leftarrow \mathcal{E}(m_{\text{query}}; \theta_{\mathcal{E}})$
4: $Z_{\text{anchors}}, W_{\text{anchors}} \leftarrow \mathcal{T}(z_{\text{query}}, Z_{\text{train}}; \theta_{\mathcal{T}})$
5: $\hat{y}_{\text{query}} \leftarrow \mathcal{P}(z_{\text{query}}, Z_{\text{anchors}}, W_{\text{anchors}}; \theta_{\mathcal{P}})$

6: **return** Predictions $\hat{y}_{\text{query}}$

---

# D   MODEL STRUCTURE

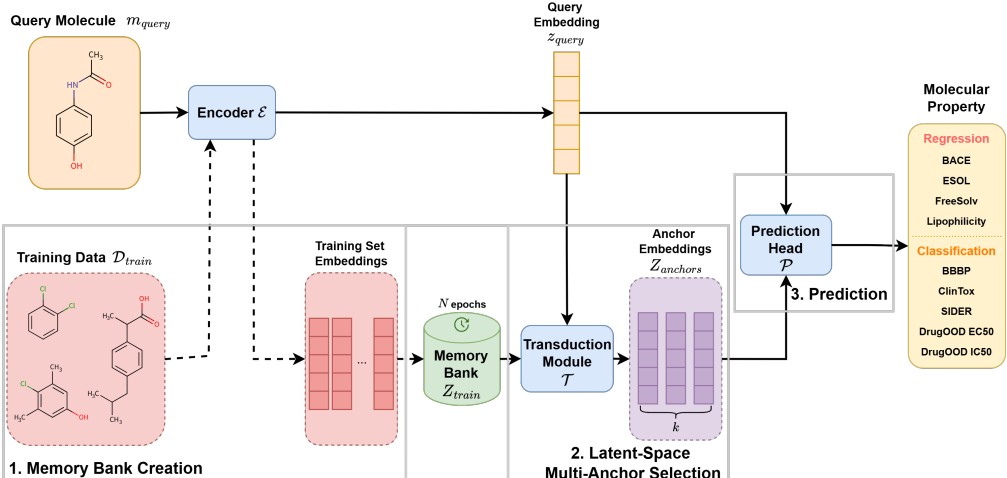

Figure 3: Overview of our proposed latent-space multi-anchor transductive framework

# E   ANCHOR SELECTION METHODS

To investigate the importance and impact of different anchor selection strategies within our transductive learning framework, we evaluated several approaches. These methods primarily differ in how they identify relevant anchors from the Train, balancing factors such as similarity to the query, anchor diversity, or adaptation to local data density. Key approaches considered include selecting the straightforward Top-K most similar anchors, methods that aim for a diverse set of anchors, adaptive selection techniques, and temperature-based sampling which introduces stochasticity. The choice of distance metric, such as Euclidean distance or cosine similarity, also plays a significant role within these strategies.

We explored the following strategies:

- **Top-k:** Selects the $k$ anchors closest (or most similar) to the query embedding.
- **Adaptive Selection:** Dynamically adjusts the number of selected anchors $k_{\text{adaptive}}$ (within predefined bounds) based on the estimated local density of training samples around the query embedding in the latent space. Anchors beyond this adaptive count might be masked or ignored in subsequent steps.
- **Temperature Sampling:** Samples $k$ anchors based on a probability distribution derived from the latent space similarities (or distances) to the query. The distribution is sharpened or softened by a temperature parameter $\tau$; lower temperatures approximate Top-k selection, while higher temperatures increase the probability of selecting less similar anchors, promoting randomness.

Figure 4 provides a comparative visualization of several of these anchor selection strategies, specifically focusing on their performance on various benchmark datasets.

The radar plots in Figure 4 illustrate these relative performances. Across the comprehensive set of evaluations detailed in this section, and considering factors such as performance consistency, robustness, and simplicity, the Top-K strategy utilizing Euclidean distance emerged as a strong and reliable default choice for our experiments.

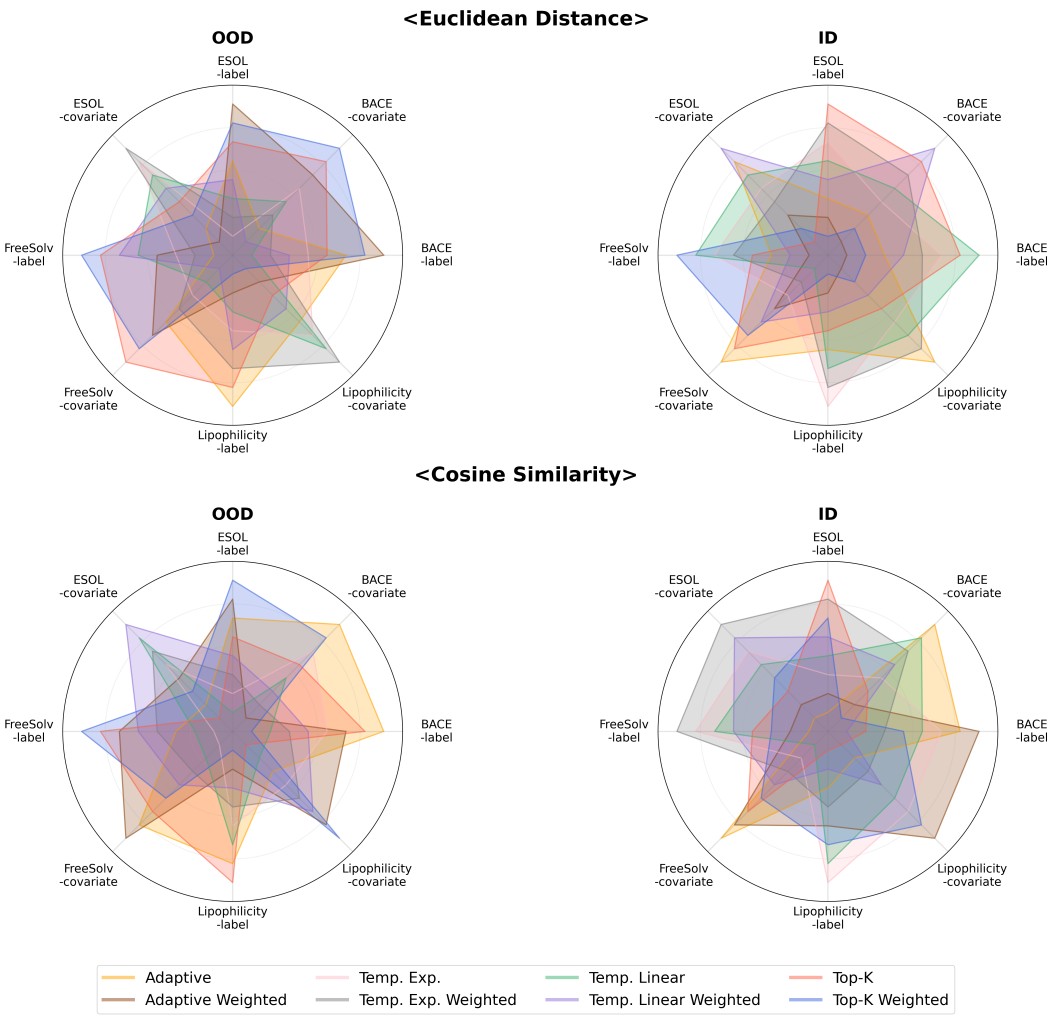

Figure 4: Performance (MAE) of anchor selection strategies by rank, depicting (a) Euclidean-based metrics and (b) Cosine-based similarity/distance metrics based methods. Each axis corresponds to a specific benchmark dataset).

# F  SPLIT METHODS

The method used to split data into Train, Test_ID, and Test_OOD sets is crucial for rigorously evaluating a model's generalization capabilities. As visualized in Figure 5, a Multidimensional Scaling (MDS) projection based on pairwise Tanimoto similarity between molecular fingerprints initially suggests that a fingerprint-based split yields a more distinct separation between train and OOD samples compared to a scaffold-based split. However, this apparent clarity can be misleading, fingerprint-level similarity does not inherently capture or reflect scaffold-level dissimilarity, which is often a more relevant measure of structural novelty in drug discovery and molecular design.

To more directly and robustly assess the introduction of structural novelty in the OOD set, we evaluated the similarity between Test_OOD and train molecules using three distinct Bemis-Murcko(BM) scaffold-level similarity metrics. The results, presented in Figure 6, demonstrate the comparative efficacy of scaffold-based versus fingerprint-based splitting strategies:

- **Scaffold Tanimoto Similarity (Figure 6a):** When assessed using Tanimoto similarity at the BM scaffold level, the scaffold-based split consistently produces a Test_OOD set with lower similarity to the Train. This indicates a clear introduction of structurally dissimilar scaffolds in the Test_OOD under this splitting regime.

- **Scaffold Maximum Common Substructure (MCS) Similarity (Figure 6b):** The trend continues with MCS similarity, a more stringent measure of structural overlap. Scaffold splits again result in Test_OOD samples that have lower MCS similarity to Train set, an effect that is particularly evident in the ESOL and FREESOLV datasets.

- **Scaffold CATS Pharmacophore Similarity (Figure 6c):** Using CATS pharmacophore similarity, which captures 3D pharmacophoric features of the scaffolds, scaffold splits generally tend to lower the functional similarity of the Test_OOD set. However, this effect is less pronounced and shows more variability across the different datasets compared to the Tanimoto and MCS metrics.

Collectively, these analyses, particularly the significant reductions in similarity observed with the Tanimoto and MCS metrics (Figure 6a and 6b), validate our adoption of scaffold-based splitting. This approach provides a more rigorous and principled methodology for creating Test_OOD sets with genuine structural novelty, which is essential for evaluating the true generalization capabilities of models in structure-based settings.

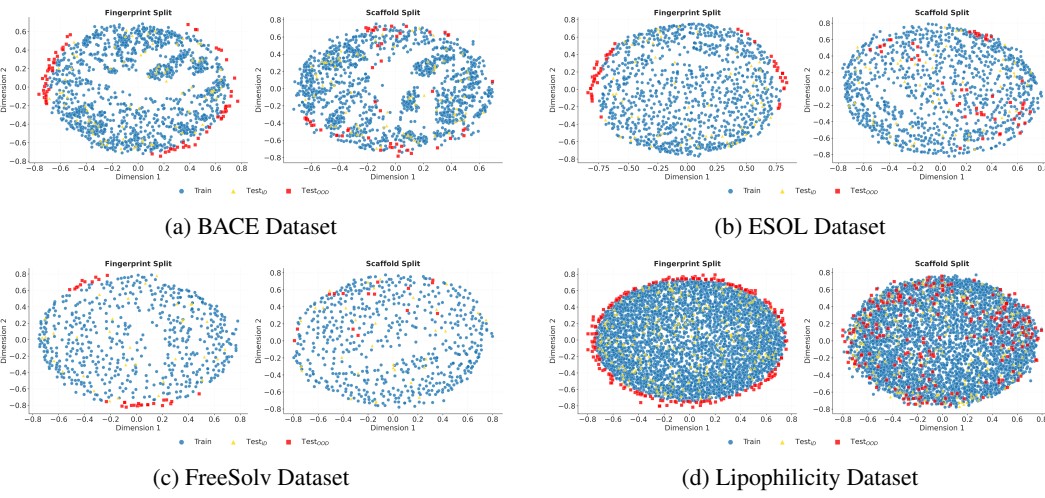

(a) BACE Dataset  (b) ESOL Dataset

(c) FreeSolv Dataset  (d) Lipophilicity Dataset

Figure 5: Chemical space distribution comparison for fingerprint and scaffold splits across four molecular property datasets. Each panel displays a 2D MDS projection based on molecular fingerprint Tanimoto similarity. Colors distinguish Train, Test_ID, and Test_OOD samples.

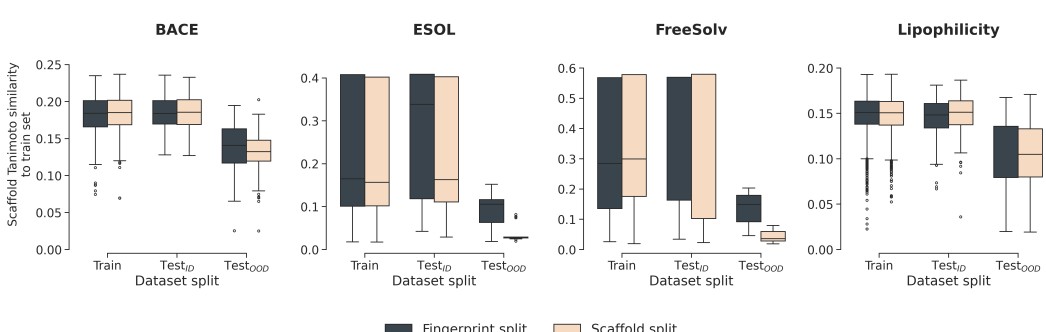

(a) Bemis-Murcko scaffold Tanimoto similarity.

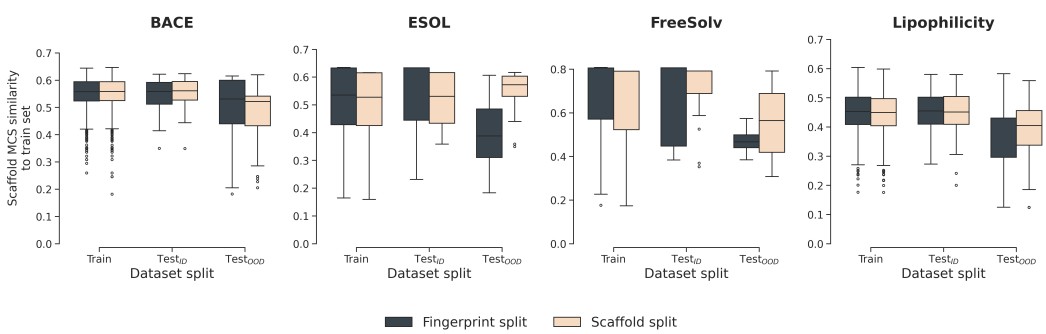

(b) Bemis-Murcko scaffold Maximum Common Substructure (MCS) similarity.

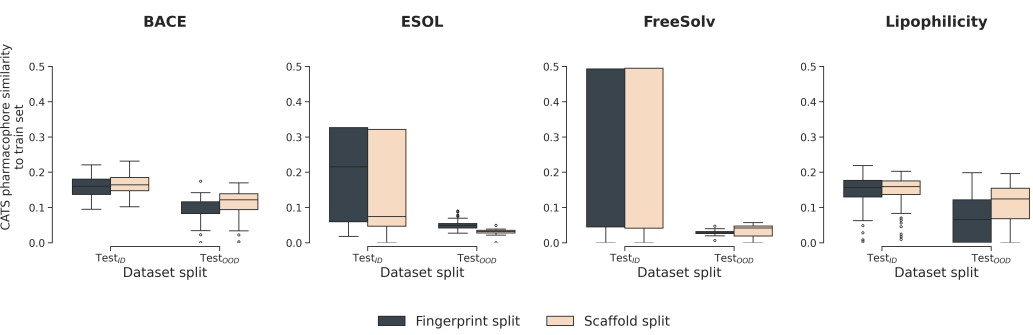

(c) Bemis-Murcko scaffold CATS pharmacophore similarity.

Figure 6: Comparison of scaffold-level similarity between the test sets and the training set for fingerprint VS scaffold splits, evaluated using three Bemis-Murcko scaffold similarity criteria: (a) Tanimoto similarity, (b) Maximum Common Substructure (MCS) similarity, and (c) CATS pharmacophore similarity. Distributions show similarity values for Training, $Test_{ID}$ (scaffold), $Test_{OOD}$ (scaffold), $Test_{ID}$ (fingerprint), and $Test_{OOD}$ (fingerprint) sets. Scaffold splits generally yield lower similarity for the $Test_{OOD}$ set, indicating stronger structural and functional distributional shifts.

# G    IMPLEMENTATION DETAILS

We trained all models on 4 * AMD EPYC 7742 64-Core Processor (256 cores) CPUs, 8 * RTX 3090 GPUs, 512GB RAM. We set 500 epochs as default and 100 epochs for large models(SMI-TED and Unimol). Hyperparameter configurations and training loss curves are presented in Table 7 and Figure 7. Checkpoints are collected at the final epoch, after convergence.

Table 7: Hyperparameter configurations for experimental setup.

| Parameters | Settings | Values |
|---|---|---|
| Batch Size | | 64, 128, 256, 512 |
| Epochs | Inductive Models | 10, 30, 50, 100 |
| | BLT, MALT (regression) | 100, 200, 500, 1000 |
| | BLT (classification) | 100, 500, 1000, 2000 |
| | MALT (classification) | 10, 20, 50, 100 |
| Number of anchors $k$ | | 1, 3, 5, 10 |
| Learning Rate | | $10^{-3}, 10^{-4}, 10^{-5}$ |

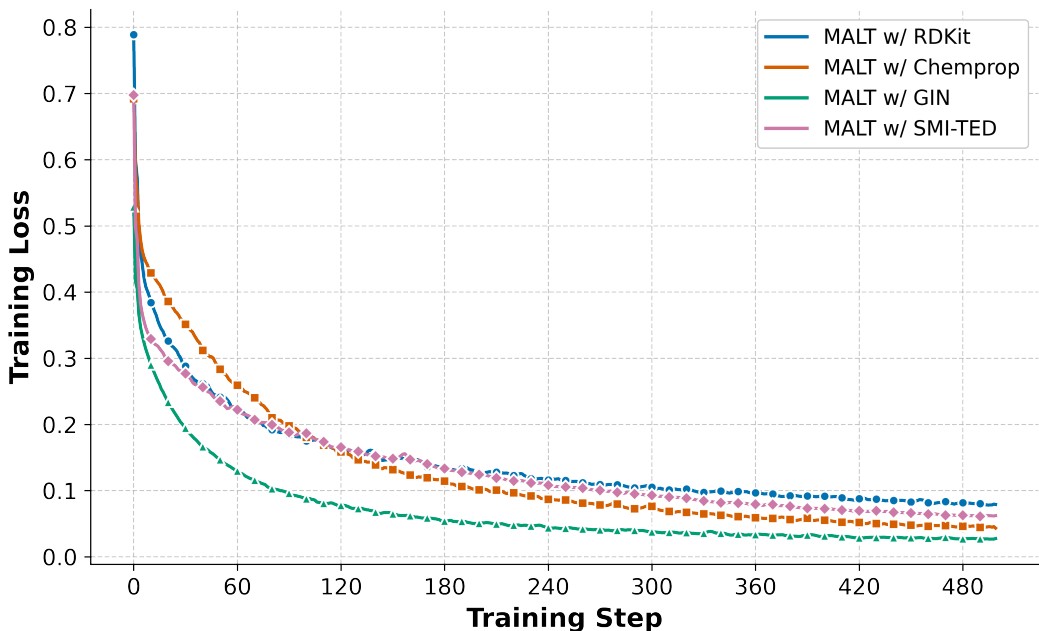

Figure 7: Comparison of training loss curves for MALT variants using RDKit, Chemprop, GIN, and SMI-TED representations across 500 training steps.

Table 8: Performance (MAE) Comparison of Encoder Training Strategies. Strategies involve inductive finetuning ("Finetune O/X") and whether the encoder is adapted during transduction ("Adapt O") or kept frozen ("Adapt X"). Cell colors in orange indicate settings where "Finetune O, Adapt O" shows lower MAE. Green indicates when other settings result in lower MAE. Bold: lowest MAE in column.

| Setting | Models | BACE $Test_{ID}$ | BACE $Test_{OOD}$ | ESOL $Test_{ID}$ | ESOL $Test_{OOD}$ | FreeSolv $Test_{ID}$ | FreeSolv $Test_{OOD}$ | Lipophilicity $Test_{ID}$ | Lipophilicity $Test_{OOD}$ |
|---|---|---|---|---|---|---|---|---|---|
| **Covariate Shift (X-Splits)** | | | | | | | | | |
| Finetune X, Adapt X | Chemprop | 0.4508 ±0.0321 | 1.0973 ±0.1283 | 0.2995 ±0.0265 | 0.6525 ±0.0569 | 0.2145 ±0.0250 | 0.5827 ±0.1009 | 0.5273 ±0.0133 | 0.8090 ±0.0330 |
| | Pretrained GNN | 0.3212 ±0.0074 | 0.6763 ±0.0624 | 0.2612 ±0.0082 | 0.5326 ±0.0031 | 0.2120 ±0.0047 | 0.4389 ±0.0380 | 0.3648 ±0.0023 | 0.5606 ±0.0102 |
| | SMI-TED light | 0.3767 ±0.0171 | 0.8553 ±0.0431 | 0.3187 ±0.0182 | 0.5508 ±0.0156 | 0.2878 ±0.0204 | 0.5210 ±0.0461 | 0.4036 ±0.0076 | 0.5584 ±0.0160 |
| Finetune O, Adapt X | Chemprop | 0.3944 ±0.0183 | 0.9481 ±0.0341 | **0.2061** ±0.0183 | **0.4676** ±0.0423 | **0.1487** ±0.0124 | 0.3423 ±0.0498 | 0.3758 ±0.0160 | 0.5545 ±0.0199 |
| | Pretrained GNN | **0.2991** ±0.0097 | 0.6810 ±0.0712 | **0.2101** ±0.0116 | 0.5310 ±0.0113 | **0.1868** ±0.0103 | 0.3694 ±0.0362 | 0.3517 ±0.0100 | 0.5478 ±0.0064 |
| | SMI-TED light | 0.3342 ±0.0192 | **0.6487** ±0.0390 | 0.2363 ±0.0115 | **0.4141** ±0.0175 | 0.1989 ±0.0076 | 0.3272 ±0.0352 | 0.4036 ±0.0076 | 0.5584 ±0.0160 |
| Finetune O, Adapt O | Chemprop | **0.2847** ±0.0165 | 0.7783 ±0.0553 | 0.2180 ±0.0049 | 0.5072 ±0.0154 | 0.1522 ±0.0043 | **0.2999** ±0.0164 | **0.3474** ±0.0135 | **0.4894** ±0.0263 |
| | Pretrained GNN | 0.3317 ±0.0083 | **0.6333** ±0.0347 | 0.2103 ±0.0113 | **0.5305** ±0.0120 | 0.1919 ±0.0126 | **0.3388** ±0.0255 | **0.3370** ±0.0007 | **0.5369** ±0.0007 |
| | SMI-TED light | **0.3037** ±0.0046 | 0.6716 ±0.0569 | **0.2057** ±0.0052 | 0.4322 ±0.0118 | **0.1497** ±0.0161 | **0.2613** ±0.0329 | **0.3608** ±0.0120 | **0.5417** ±0.0164 |
| **Label Shift (Y-Splits)** | | | | | | | | | |
| Finetune X, Adapt X | Chemprop | 0.4572 ±0.0263 | 1.0237 ±0.1289 | 0.2594 ±0.0229 | 0.5097 ±0.0229 | 0.3075 ±0.0270 | 0.3461 ±0.0525 | 0.4991 ±0.0305 | 0.9026 ±0.0708 |
| | Pretrained GNN | 0.3999 ±0.0057 | **0.7052** ±0.0254 | 0.2293 ±0.0084 | 0.5607 ±0.0134 | 0.2124 ±0.0269 | 0.3453 ±0.0156 | 0.3479 ±0.0113 | 0.5004 ±0.0068 |
| | SMI-TED light | 0.4104 ±0.0139 | 0.9685 ±0.0343 | 0.3686 ±0.0068 | 0.8628 ±0.0175 | 0.2868 ±0.0163 | 0.8585 ±0.0448 | 0.5246 ±0.0052 | 1.1654 ±0.0363 |
| Finetune O, Adapt X | Chemprop | 0.4132 ±0.0325 | 0.9366 ±0.0706 | 0.1941 ±0.0079 | **0.4173** ±0.0284 | 0.2019 ±0.0173 | **0.2009** ±0.0516 | 0.3489 ±0.0141 | **0.5797** ±0.0571 |
| | Pretrained GNN | 0.4057 ±0.0111 | 0.7464 ±0.0158 | 0.1849 ±0.0090 | 0.5000 ±0.0283 | **0.1386** ±0.0151 | 0.3516 ±0.0186 | 0.3497 ±0.0767 | 0.4903 ±0.0051 |
| | SMI-TED light | 0.3592 ±0.0092 | 0.9228 ±0.0226 | 0.2167 ±0.0140 | 0.5566 ±0.0281 | 0.2493 ±0.0245 | 0.4763 ±0.0395 | 0.3480 ±0.0113 | 0.7379 ±0.0206 |
| Finetune O, Adapt O | Chemprop | **0.3461** ±0.0097 | **0.7705** ±0.0287 | **0.1734** ±0.0070 | 0.4994 ±0.0114 | **0.1469** ±0.0179 | 0.2753 ±0.0530 | **0.3185** ±0.0104 | 0.6444 ±0.0151 |
| | Pretrained GNN | **0.3861** ±0.0097 | 0.7340 ±0.0121 | **0.1845** ±0.0093 | **0.4989** ±0.0267 | 0.1585 ±0.0165 | **0.2637** ±0.0220 | **0.3195** ±0.0108 | **0.4690** ±0.0108 |
| | SMI-TED light | **0.3546** ±0.0164 | **0.8326** ±0.0141 | **0.1852** ±0.0096 | **0.5390** ±0.0199 | **0.1716** ±0.0061 | **0.2856** ±0.0427 | **0.3300** ±0.0097 | **0.6609** ±0.0331 |

Table 9: Performance(MAE) Comparison of Top $k$ Anchor Selection Strategies. Best results bolded, second best underlined. Performance shown across varying $k$ values.

| Anchor Strategy | Covariate Shift (X-Splits) $Test_{OOD}$ BACE | Esol | FreeSolv | Lipo | Covariate Shift $Test_{ID}$ BACE | Esol | FreeSolv | Lipo | Label Shift (Y-Splits) $Test_{OOD}$ BACE | Esol | FreeSolv | Lipo | Label Shift $Test_{ID}$ BACE | Esol | FreeSolv | Lipo |
|---|---|---|---|---|---|---|---|---|---|---|---|---|---|---|---|---|
| *Top k* | | | | | | | | | | | | | | | | |
| k = 1 | 0.6446 | 0.5804 | 0.4210 | 0.5409 | 0.3464 | 0.2658 | 0.2154 | 0.3724 | _0.7510_ | 0.5424 | 0.4001 | 0.5860 | _0.3886_ | 0.2397 | 0.1697 | 0.3398 |
| k = 3 | 0.6933 | 0.5652 | 0.4031 | 0.5343 | _0.3171_ | 0.2061 | _0.1906_ | _0.3402_ | 0.7587 | 0.5044 | 0.3715 | _0.4713_ | 0.4041 | 0.2154 | 0.1423 | 0.3191 |
| k = 5 | **0.6333** | 0.5515 | 0.3797 | **0.5263** | **0.3028** | 0.2151 | **0.1875** | 0.3443 | 0.7607 | 0.5040 | _0.3573_ | 0.4842 | 0.4028 | 0.1906 | 0.1421 | **0.3125** |
| *Ours (Top 10)* | **0.6333** | 0.5305 | **0.3388** | _0.5369_ | 0.3317 | **0.2103** | 0.1919 | 0.3370 | **0.7340** | **0.4989** | **0.2637** | **0.4690** | 0.3861 | **0.1845** | 0.1585 | _0.3195_ |
| *Ours ($11^{th}$ to $(10+k)^{th}$)* | | | | | | | | | | | | | | | | |
| k = 1 | 0.6675 | 0.5255 | _0.3726_ | 0.6028 | 0.3557 | 0.2327 | 0.2094 | 0.3860 | 0.8211 | 0.5068 | 0.3642 | 0.4954 | 0.4446 | 0.2017 | 0.1536 | 0.3646 |
| k = 3 | _0.6664_ | **0.5218** | 0.3809 | 0.5677 | 0.3380 | 0.2209 | 0.1985 | 0.3729 | 0.8129 | _0.4997_ | 0.3679 | 0.4903 | 0.4406 | 0.1906 | 0.1448 | 0.3494 |
| k = 5 | 0.6710 | _0.5224_ | 0.3847 | 0.5575 | 0.3326 | 0.2194 | _0.1943_ | 0.3718 | 0.8062 | 0.5032 | 0.3694 | 0.4915 | 0.4409 | _0.1887_ | _0.1412_ | 0.3418 |
| k = 10 | 0.6715 | 0.5259 | 0.3783 | _0.5516_ | 0.3237 | 0.2212 | 0.1953 | 0.3758 | 0.7896 | 0.5022 | 0.3783 | 0.4879 | 0.4345 | 0.1892 | **0.1345** | 0.3419 |

Table 10: MALT-GIN's Robustness to Noisy Anchors. Performance (MAE) is shown for $Test_{ID}$ and $Test_{OOD}$ as the number of top-ranked anchors is replaced by the lowest-ranked (noisiest) ones. The model shows stable performance, indicating the attention mechanism effectively discounts irrelevant information.

| | BACE | | ESOL | | FreeSolv | | Lipophilicity | |
|---|---|---|---|---|---|---|---|---|
| Noisy Anchors | $Test_{ID}$ | $Test_{OOD}$ | $Test_{ID}$ | $Test_{OOD}$ | $Test_{ID}$ | $Test_{OOD}$ | $Test_{ID}$ | $Test_{OOD}$ |
| **Covariate Shift (X-Splits)** | | | | | | | | |
| **0 (Default)** | 0.3338 | 0.7290 | 0.2142 | 0.5572 | 0.2085 | 0.4477 | 0.3003 | 0.4736 |
| **1** | 0.3583 | 0.6343 | 0.2281 | 0.5683 | 0.2133 | 0.4395 | 0.2992 | 0.4713 |
| **3** | 0.3338 | 0.9222 | 0.2158 | 0.6063 | 0.2206 | 0.4603 | 0.3003 | 0.4746 |
| **5** | 0.3269 | 0.8346 | 0.2186 | 0.5853 | 0.2175 | 0.4486 | 0.3009 | 0.4748 |
| **Label Shift (Y-Splits)** | | | | | | | | |
| **0 (Default)** | 0.4191 | 0.7115 | 0.2160 | 0.5395 | 0.2086 | 0.4235 | 0.3024 | 0.4743 |
| **1** | 0.4206 | 0.6879 | 0.2264 | 0.5146 | 0.2139 | 0.4074 | 0.3019 | 0.4746 |
| **3** | 0.4344 | 0.7271 | 0.2287 | 0.5279 | 0.2207 | 0.4393 | 0.3020 | 0.4736 |
| **5** | 0.4061 | 0.7068 | 0.2264 | 0.5041 | 0.2181 | 0.4298 | 0.3027 | 0.4747 |

Table 11: Comparison of MALT-GIN against k-NN averaging baselines on the OOD test set (Covariate Shift). The full MALT model outperforms all simpler retrieval-based methods. Best results are in **bold**.

| Method | BACE | ESOL | FreeSolv | Lipo |
|---|---|---|---|---|
| **MALT-GIN (Ours)** | **0.633** | **0.531** | **0.339** | **0.537** |
| *k-NN Averaging Baselines (MAE)* | | | | |
| MALT-GIN embedding | 0.726 | 0.585 | 0.618 | 0.597 |
| Pretrained GIN embedding | 0.890 | 0.695 | 0.396 | 0.871 |
| ECFP (Tanimoto) | 0.841 | 0.773 | 0.551 | 0.921 |
| Random Selection | 1.385 | 1.365 | 0.949 | 1.252 |

Table 12: OOD performance (MAE) on QM9 HOMO and LUMO prediction. Best results for each target are in **bold**.

| Model | HOMO | LUMO |
|---|---|---|
| MALT(Chemprop) | **1.9960 ± 0.0010** | **0.9900 ± 0.0184** |
| Chemprop | 2.3452 ± 0.0405 | 1.1904 ± 0.0244 |
| MALT(GIN) | **2.2502 ± 0.0723** | **1.2799 ± 0.0047** |
| Pre-trained GIN | 2.3488 ± 0.0150 | 1.4612 ± 0.0189 |

# I PARITY PLOT

To assess the effectiveness of our method across both inductive and transductive settings, we present parity plots comparing the predicted and true values on various datasets and split types. As shown in Figure 8 and Figure 9, we evaluate several inductive models (Chemprop, GIN, and SMI-TED) and their MALT-enhanced variants under the label split. Across all base architectures, the MALT-integrated models yield predictions that are more closely aligned with the ideal diagonal and achieve lower mean absolute error (MAE), demonstrating consistent performance gains. In the transductive setting, Figure 10 compares the baseline BLT model with our proposed MALT-based model across four datasets and two OOD splits (covariate and label). In most cases, our method reduces the prediction error and aligns the outputs more tightly with the ground truth, validating its generalization capability under both feature and label distribution shifts.

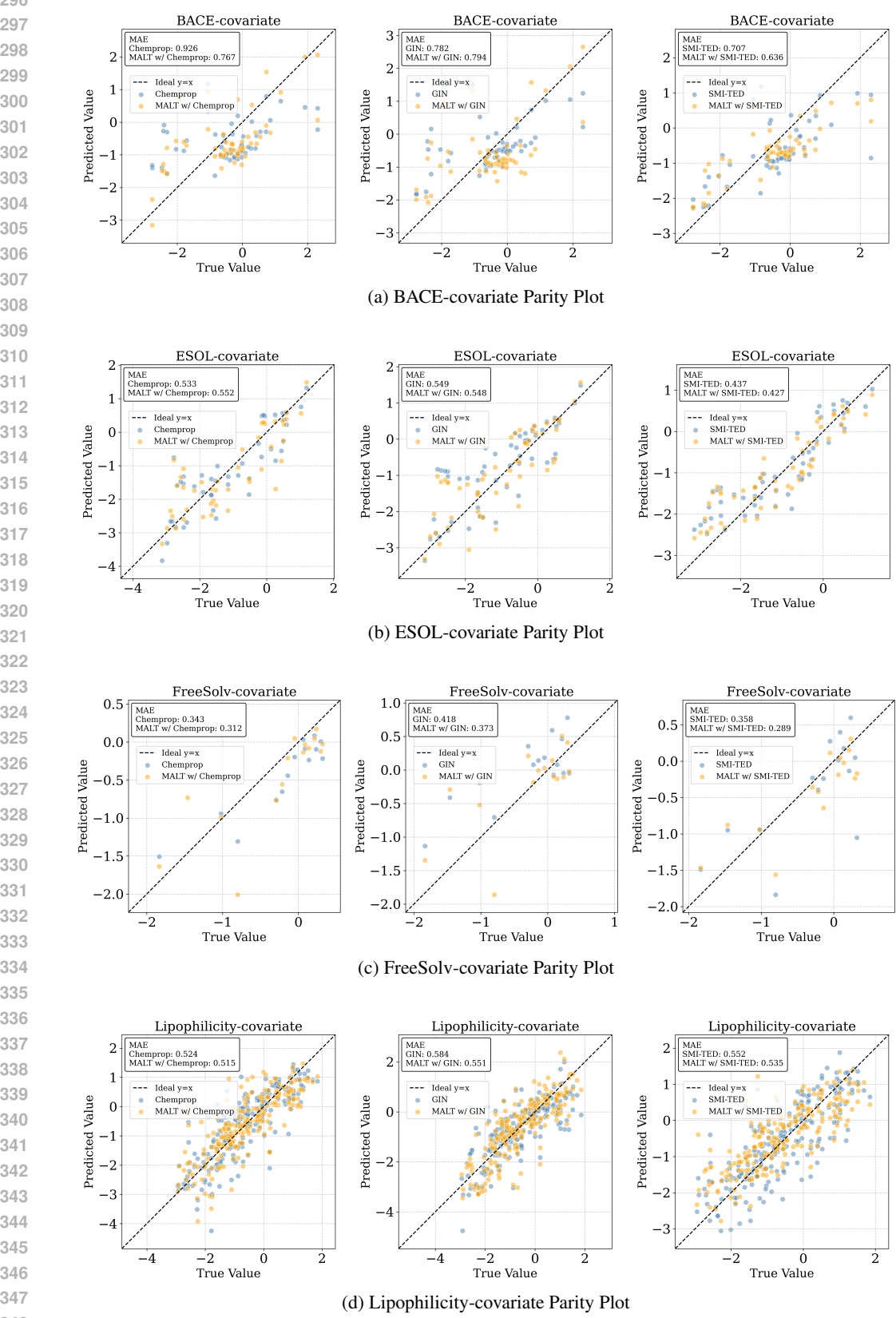

Figure 8: Parity plots comparing various inductive models and their MALT-enhanced variants under the covariate split.

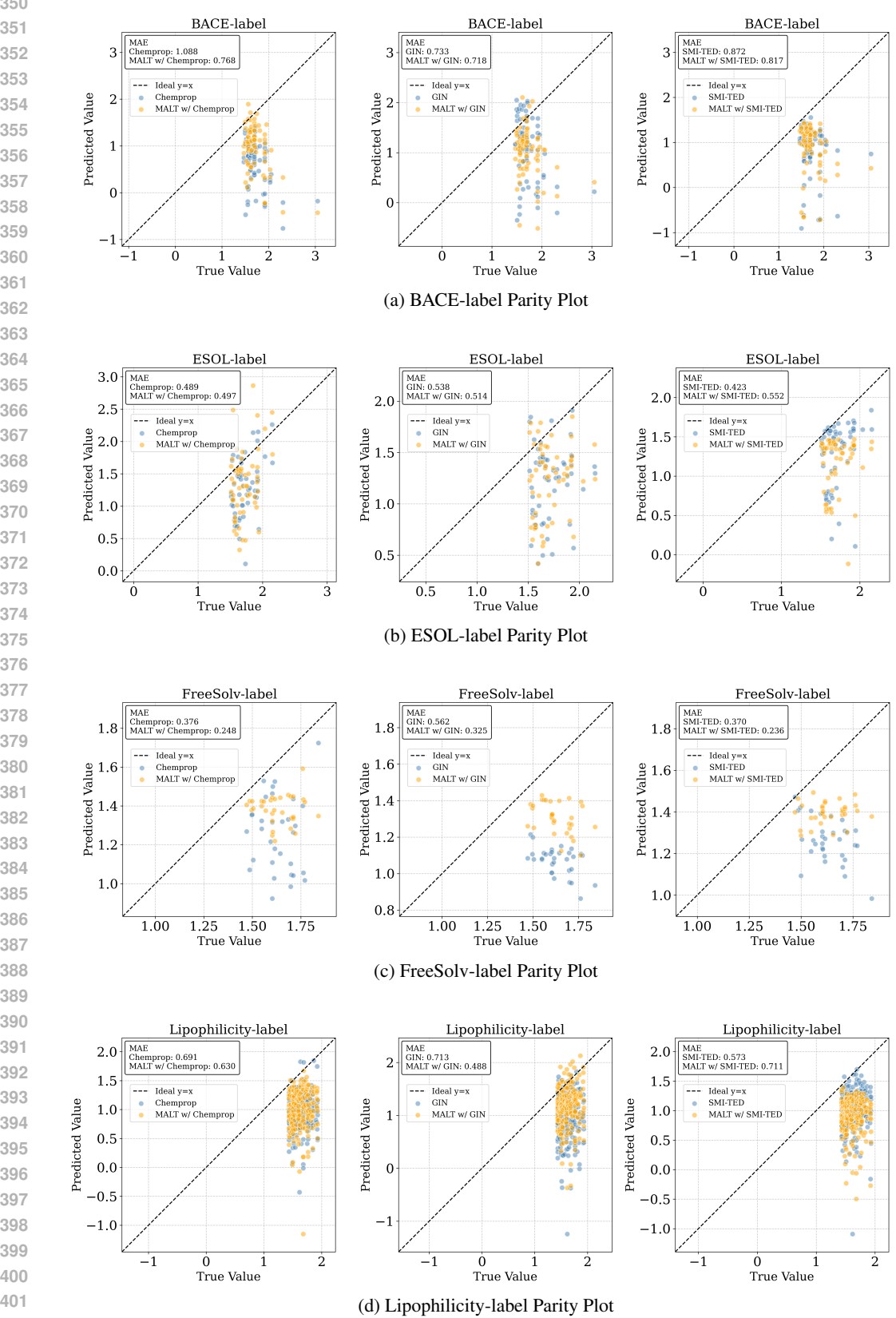

Figure 9: Parity plots comparing various inductive models and their MALT-enhanced variants under the label split.

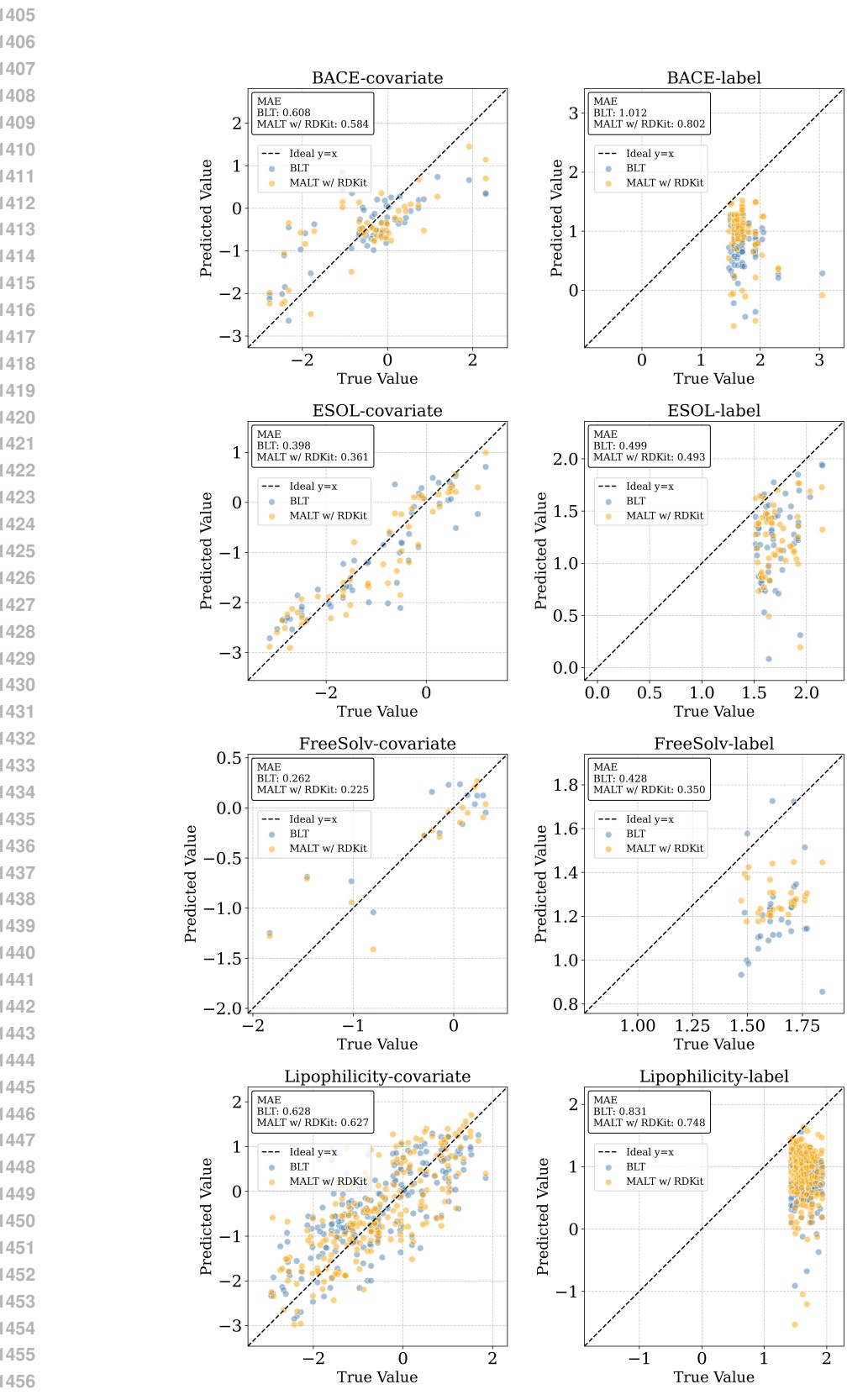

Figure 10: Parity plot comparison between BLT and MALT trained with RDkit across different datasets.

## J Embedding Space Transformation Comparison

To further understand how our model reshapes the representation space, we visualize the embedding space of molecules before and after training using t-SNE. As illustrated in Figure 11 and Figure 12, we observe that the selected query point (red star) becomes more tightly clustered with its corresponding anchor points after training. This consistent contraction across multiple datasets suggests that the model effectively aligns semantically similar molecules in the latent space, promoting smoother generalization to OOD queries.

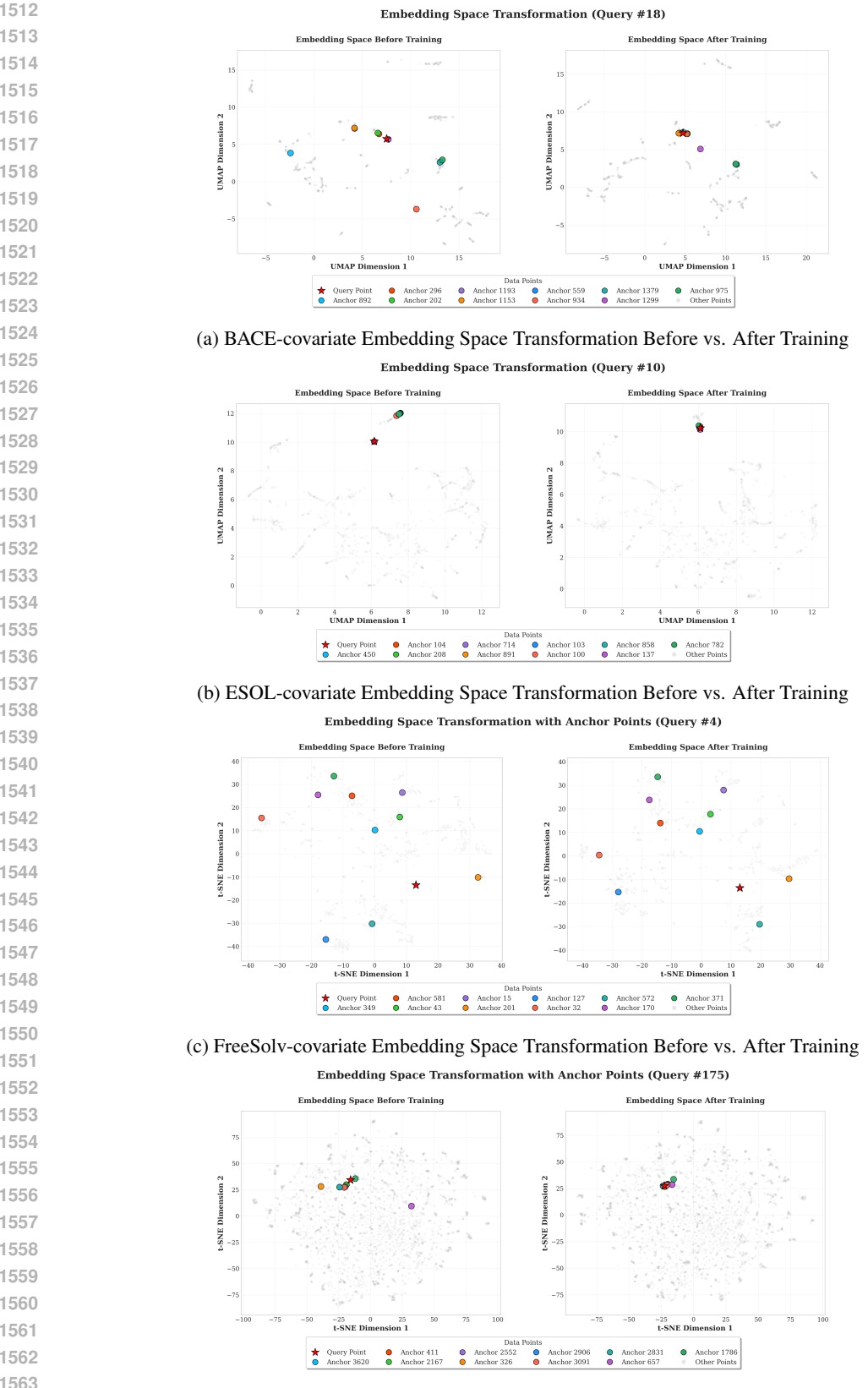

(a) BACE-covariate Embedding Space Transformation Before vs. After Training

(b) ESOL-covariate Embedding Space Transformation Before vs. After Training

(c) FreeSolv-covariate Embedding Space Transformation Before vs. After Training

(d) Lipophilicity-covariate Embedding Space Transformation Before vs. After Training

Figure 11: Embedding space transformation from selected anchor embeddings before and after MALT training with GIN under the covariate split.

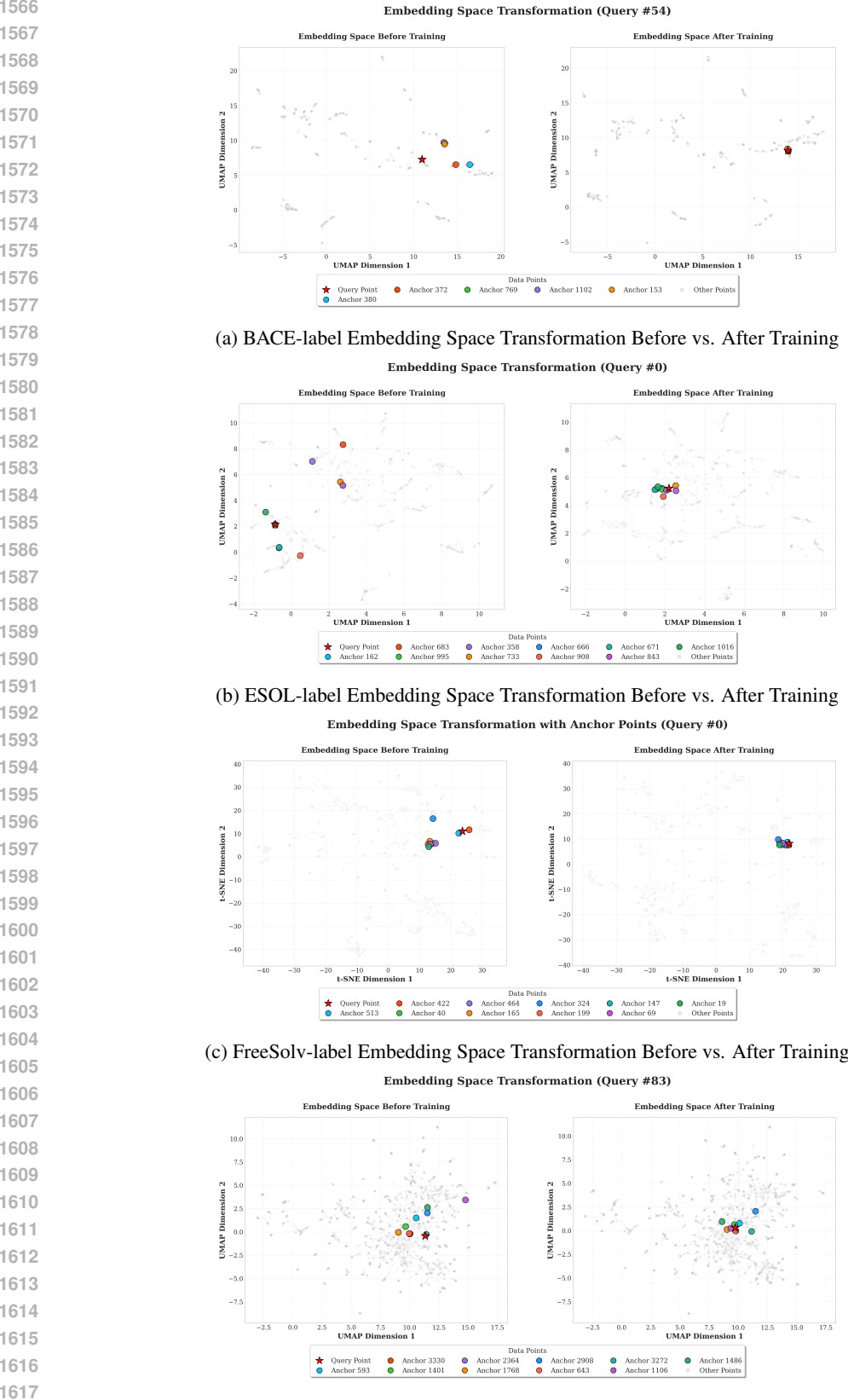

(a) BACE-label Embedding Space Transformation Before vs. After Training

(b) ESOL-label Embedding Space Transformation Before vs. After Training

(c) FreeSolv-label Embedding Space Transformation Before vs. After Training

(d) Lipophilicity-label Embedding Space Transformation Before vs. After Training

Figure 12: Embedding space transformation from selected anchor embeddings before and after MALT training with GIN under the label split.

# K  THEORETICAL ANALYSIS OF MULTI-ANCHOR FUSION

We analyze why fusing multiple anchors improves OOD generalization. Working in a latent space $\mathcal{Z}$ induced by a frozen encoder $E : \mathcal{X} \to \mathcal{Z}$, subtraction is well-defined so we can apply BLT in $(\Delta z, z')$ as in (11). (BLT Assumptions 3.1–3.3 and Thm. 1 are stated over $(\Delta x, x')$; with $z = E(x)$ the same setup holds over $(\Delta z, z')$ under the same assumptions.)

**Setup.** For a query $x$ with $z = E(x)$ and a set of anchors $S(x) \subseteq \mathcal{Z}$, define the single–anchor predictor
$$h_\theta^{(z')}(x) := \langle f_\theta(z - z'), g_\theta(z') \rangle, \qquad z' \in S(x),$$
and the fused predictor
$$H_\Theta(x) := \sum_{z' \in S(x)} \alpha(x, z')\, h_\theta^{(z')}(x), \quad \alpha(x, z') \geq 0, \ \sum_{z' \in S(x)} \alpha(x, z') = 1.$$

Write $\alpha_j = \alpha(x, z^{(j)})$, $h^{(j)} = h_\theta^{(z^{(j)})}$, and let $y_\star(x)$ be the scalar target. Define $\varepsilon_j(x) = h^{(j)}(x) - y_\star(x)$, its conditional mean $\mu_j(x) = \mathbb{E}[\varepsilon_j(x) \mid x]$, and centered part $\eta_j(x) = \varepsilon_j(x) - \mu_j(x)$. We also use the anchor-indexed error $\varepsilon(x, z') := h_\theta^{(z')}(x) - y_\star(x)$.

**Anchor mixture at test time.** An anchor selection rule is a conditional distribution $Q(\cdot \mid x)$ supported on $S(x)$, which induces latent joint distributions
$$\overline{D}_{\text{train}} : (\Delta z, z') \quad \text{and} \quad \overline{D}_{\text{test}}^Q : (\Delta z, z') \text{ with } z' \sim Q(\cdot \mid x), \ \Delta z = z - z'.$$
We assume BLT Assumptions 3.1–3.3 hold in latent space for $(\overline{D}_{\text{train}}, \overline{D}_{\text{test}}^Q)$.

**Effective number of anchors and correlation control.** For fixed $x$,
$$k_{\text{eff}}(x) := \frac{1}{\sum_j \alpha_j^2} \in [1, |S(x)|], \qquad k_{\text{eff}}^{\min} := \inf_x k_{\text{eff}}(x).$$
Assume bounded conditional cross–correlation among centered errors:
$$\big|\mathrm{Corr}(\eta_j(x), \eta_\ell(x) \mid x)\big| \leq \rho \in [0, 1) \quad \text{for all } j \neq \ell, \tag{6}$$
with the convention that $\mathrm{Corr} = 0$ if either conditional variance is zero.

**Lemma K.1** (Variance bound under bounded cross–correlation)**.** *For any fixed $x$,*
$$\mathbb{E}\left[\left(\sum_j \alpha_j\, \eta_j(x)\right)^2 \,\Big|\, x\right] \leq \left(\tfrac{1-\rho}{k_{\text{eff}}(x)} + \rho\right) \cdot \max_j \mathrm{Var}(\eta_j(x) \mid x).$$

*Proof.* Let $\Sigma_{j\ell} = \mathrm{Cov}(\eta_j, \eta_\ell \mid x)$ and $\sigma_{\max}^2 = \max_j \Sigma_{jj}$. By equation 6, $|\Sigma_{j\ell}| \leq \rho \sqrt{\Sigma_{jj}\Sigma_{\ell\ell}} \leq \rho\, \sigma_{\max}^2$ for $j \neq \ell$. Hence
$$\alpha^\top \Sigma \alpha \leq \sigma_{\max}^2 \left(\sum_j \alpha_j^2 + \rho \sum_{j \neq \ell} \alpha_j \alpha_\ell\right) = \sigma_{\max}^2 \left((1 - \rho)\sum_j \alpha_j^2 + \rho\right) = \sigma_{\max}^2 \left(\tfrac{1-\rho}{k_{\text{eff}}(x)} + \rho\right).$$
$\square$

**Theorem K.2** (Conditional MSE decomposition for fusion)**.** *Under equation 6 and convex weights,*
$$\mathbb{E}\left[\left(H_\Theta(x) - y_\star(x)\right)^2 \,\Big|\, x\right] \leq \underbrace{\left(\sum_j \alpha_j \mu_j(x)\right)^2}_{\textit{squared bias}} + \left(\tfrac{1-\rho}{k_{\text{eff}}(x)} + \rho\right) \cdot \max_j \mathrm{Var}(\eta_j(x) \mid x).$$

*Consequently,*
$$R_{\text{test}}(H_\Theta) \leq \mathbb{E}\left[\left(\sum_j \alpha_j \mu_j(x)\right)^2\right] + \left(\tfrac{1-\rho}{k_{\text{eff}}^{\min}} + \rho\right) \cdot \mathbb{E}\left[\max_j \mathrm{Var}(\eta_j(x) \mid x)\right].$$
*If $\mu_j(x) \equiv 0$ (per–anchor calibration), the multiplicative improvement on the variance term is $\tfrac{1-\rho}{k_{\text{eff}}^{\min}} + \rho \leq 1$, with equality only if $k_{\text{eff}}^{\min} = 1$ or $\rho = 1$.*

*Proof.* Write $H_\Theta - y_\star = \sum_j \alpha_j \varepsilon_j = \sum_j \alpha_j \mu_j + \sum_j \alpha_j \eta_j$ and apply Lemma K.1 to the centered part. For the population bound, note $k_{\text{eff}}(x) \geq k_{\text{eff}}^{\min}$ and take expectations over $x$. $\square$

**Connection to BLT.** Let $Q_\alpha(\cdot \mid x)$ be the distribution on $S(x)$ with density $\alpha(x, \cdot)$. Assume BLT Assumptions 3.1–3.3 hold in latent space for $(\overline{D}_{\text{train}}, \overline{D}_{\text{test}}^{Q_\alpha})$ so that BLT Thm. 1 applies (11).

**Theorem K.3** (BLT-on-mixture bound for $H_\Theta$). *With squared loss and Assumption above,*

$$R_{\text{test}}(H_\Theta) \leq \mathbb{E}_{(x,z')\sim\overline{D}_{\text{test}}^{Q_\alpha}}\big[(h_\theta^{(z')}(x) - y_\star(x))^2\big] \leq C_{\text{BLT}} \cdot R_{\text{train}}, \qquad C_{\text{BLT}} = \text{poly}(\kappa, M/\sigma).$$

*Proof.* For fixed $x$, Jensen yields $(H_\Theta(x) - y_\star(x))^2 \leq \sum_{z'} \alpha(x, z') \, (h_\theta^{(z')}(x) - y_\star(x))^2$. Taking expectations gives the first inequality; BLT Thm. 1 gives the second. □

**Putting it together.** Combining Theorems K.2 and K.3,

$$R_{\text{test}}(H_\Theta) \leq \min\Big\{ C_{\text{BLT}} \, R_{\text{train}}, \; \mathbb{E}\Big[\big(\textstyle\sum_j \alpha_j \, \mu_j(x)\big)^2\Big] + \Big(\tfrac{1-\rho}{k_{\text{eff}}^{\min}} + \rho\Big) \mathbb{E}\Big[\max_j \text{Var}(\eta_j(x) \mid x)\Big]\Big\}.$$

If, in addition, $\text{Var}(\eta_j(x) \mid x) \leq \sigma^2$ uniformly over $x, j$, the last expectation can be replaced by $\sigma^2$.

## L  FURTHER EXPERIMENTS ACROSS PRACTICAL DRUG DISCOVERY SCENARIOS

To further address the issue that standard scaffold splits may not fully capture the complexity of real-world distributional shifts, we evaluated our framework on two more practical and chemically meaningful OOD scenarios: activity cliffs and the Lo-Hi drug discovery benchmark.

### L.1  PERFORMANCE ON ACTIVITY CLIFFS BENCHMARK

We augmented our evaluation with an **activity cliffs** benchmark, a difficult OOD challenge where structurally similar compounds exhibit large differences in potency. Using the dataset from (28), we defined the OOD test set as molecule pairs with high structural similarity but at least a tenfold difference in potency.

Across 30 pharmacological endpoints, our MALT framework demonstrated superior performance. As summarized in Table 13, MALT-enhanced models achieved a top-2 rank far more frequently than their base counterparts on both in-distribution and OOD(activity cliff) data, resulting in substantial median RMSE reductions. The detailed per-dataset results are presented in Table 14 and Table 15.

Table 13: Comprehensive performance summary on the activity cliffs benchmark across 29 datasets. MALT significantly increases the number of top-2 finishes and reduces the median RMSE compared to its base models on both ID and OOD splits.

| Model | Test Split | MALT Top-2 | Base Top-2 | Median RMSE Reduction (%) |
|---|---|---|---|---|
| **GIN** | In-Distribution | 2 | 1 | **8.1%** |
| | OOD | 18 | 2 | **12.7%** |
| **Chemprop** | In-Distribution | 11 | 0 | **8.7%** |
| | OOD | 16 | 3 | **7.1%** |
| **UniMol** | In-Distribution | 28 | 16 | **7.9%** |
| | OOD | 14 | 5 | **3.9%** |

### L.2  PERFORMANCE ON LO-HI DRUG DISCOVERY BENCHMARK

We further tested our model on the Lo-Hi benchmark (30), which simulates two distinct stages of a drug discovery campaign: Hit Identification (HI) and Lead Optimization (LO). Hi is about identifying novel, patentable drug-like molecules far from the training set, testing a model's generalization. Lo is about optimizing known hits by predicting effects of small modifications, testing a model's fine-grained sensitivity. As shown in Table 16, MALT-GIN consistently outperforms baseline GIN model in LO splits, demonstrating performance gains.

Table 14: Detailed In-Distribution Results (RMSE) for the Activity Cliffs Benchmark. The best result is in **bold** and the second-best is underlined.

| Dataset | MALT(GIN) | GIN | MALT(Chemprop) | Chemprop | MALT_UniMol | UniMol |
|---|---|---|---|---|---|---|
| CHEMBL1862_Ki | 0.837 | 0.840 | 0.852 | 0.961 | **0.707** | 0.800 |
| CHEMBL1871_Ki | 0.676 | 0.655 | 0.634 | 0.646 | **0.535** | 0.652 |
| CHEMBL2034_Ki | **0.586** | 0.839 | 0.780 | 0.809 | 0.637 | 0.713 |
| CHEMBL2047_EC50 | 0.760 | 0.705 | 0.633 | 0.794 | **0.616** | 0.715 |
| CHEMBL204_Ki | 0.739 | 0.891 | 0.709 | 0.859 | **0.629** | 0.720 |
| CHEMBL2147_Ki | 0.842 | 1.008 | 0.688 | 0.801 | **0.567** | 0.609 |
| CHEMBL214_Ki | 0.663 | 0.777 | 0.576 | 0.660 | **0.563** | 0.614 |
| CHEMBL218_EC50 | 0.770 | 0.816 | 0.811 | 0.823 | **0.644** | 0.699 |
| CHEMBL219_Ki | 0.799 | 0.826 | 0.692 | 0.784 | **0.656** | 0.706 |
| CHEMBL228_Ki | 0.730 | 0.843 | 0.719 | 0.764 | **0.664** | 0.678 |
| CHEMBL231_Ki | 0.722 | 0.788 | 0.734 | 0.812 | **0.610** | 0.699 |
| CHEMBL233_Ki | 0.842 | 0.950 | 0.770 | 0.837 | **0.736** | 0.772 |
| CHEMBL234_Ki | 0.753 | 0.845 | 0.718 | 0.768 | **0.646** | 0.700 |
| CHEMBL235_EC50 | 0.705 | 0.767 | 0.626 | 0.705 | **0.546** | 0.610 |
| CHEMBL236_Ki | 0.791 | 0.849 | 0.754 | 0.781 | **0.697** | 0.744 |
| CHEMBL237_EC50 | 0.842 | 0.905 | **0.666** | 0.976 | 0.777 | 0.910 |
| CHEMBL237_Ki | 0.728 | 0.854 | **0.667** | 0.728 | 0.674 | 0.701 |
| CHEMBL238_Ki | 0.648 | 0.696 | 0.630 | 0.738 | **0.559** | 0.622 |
| CHEMBL239_EC50 | 0.695 | 0.679 | 0.639 | 0.685 | **0.625** | 0.709 |
| CHEMBL244_Ki | 0.711 | 0.944 | 0.717 | 0.807 | **0.669** | 0.694 |
| CHEMBL262_Ki | 0.810 | 0.804 | 0.838 | 0.946 | **0.697** | 0.761 |
| CHEMBL264_Ki | 0.652 | 0.725 | 0.580 | 0.635 | **0.541** | 0.564 |
| CHEMBL2835_Ki | 0.426 | 0.462 | **0.339** | 0.433 | 0.462 | 0.417 |
| CHEMBL287_Ki | 0.791 | 0.809 | 0.761 | 0.793 | **0.660** | 0.697 |
| CHEMBL2971_Ki | 0.696 | 0.785 | 0.662 | 0.719 | **0.609** | 0.649 |
| CHEMBL3979_EC50 | **0.598** | 0.726 | 0.660 | 0.715 | 0.621 | 0.654 |
| CHEMBL4005_Ki | 0.648 | 0.780 | 0.638 | 0.698 | **0.492** | 0.569 |
| CHEMBL4203_Ki | 0.883 | 0.849 | 0.918 | 0.952 | **0.825** | 0.888 |
| CHEMBL4792_Ki | 0.882 | 0.751 | 0.678 | 0.792 | **0.614** | 0.681 |

Table 15: Detailed OOD Results (RMSE) for the Activity Cliffs Benchmark. The best result is in **bold** and the second-best is underlined.

| Dataset | MALT(GIN) | GIN | MALT(Chemprop) | Chemprop | MALT_UniMol | UniMol |
|---|---|---|---|---|---|---|
| CHEMBL1862_Ki | **0.765** | 1.041 | 0.811 | 0.846 | 0.782 | 0.833 |
| CHEMBL1871_Ki | 1.064 | **0.859** | 0.909 | 0.984 | 0.928 | 0.980 |
| CHEMBL2034_Ki | 0.836 | 1.045 | 0.940 | 0.952 | **0.827** | 0.843 |
| CHEMBL2047_EC50 | 0.827 | 0.694 | **0.625** | 0.772 | 0.838 | 0.756 |
| CHEMBL204_Ki | 0.905 | 1.128 | 0.954 | 1.074 | **0.853** | 0.916 |
| CHEMBL2147_Ki | **0.635** | 1.182 | 0.658 | 0.836 | 0.649 | 0.708 |
| CHEMBL214_Ki | **0.771** | 0.883 | 0.775 | 0.796 | 0.797 | 0.797 |
| CHEMBL218_EC50 | 0.802 | 0.853 | 0.790 | 0.849 | **0.733** | 0.813 |
| CHEMBL219_Ki | **0.757** | 0.863 | 0.775 | 0.821 | 0.840 | 0.874 |
| CHEMBL228_Ki | **0.727** | 0.887 | 0.729 | 0.881 | 0.846 | 0.806 |
| CHEMBL231_Ki | 0.982 | 1.034 | 0.908 | **0.833** | 0.923 | 0.861 |
| CHEMBL233_Ki | **0.878** | 0.995 | 0.883 | 0.950 | 0.890 | 0.903 |
| CHEMBL234_Ki | 0.734 | 0.875 | **0.619** | 0.707 | 0.669 | 0.701 |
| CHEMBL235_EC50 | 0.830 | 0.889 | **0.770** | 0.838 | 0.814 | 0.859 |
| CHEMBL236_Ki | 0.855 | 0.936 | 0.885 | 0.924 | **0.799** | 0.849 |
| CHEMBL237_EC50 | **0.905** | 0.972 | 0.940 | 0.999 | 0.914 | 1.014 |
| CHEMBL237_Ki | 0.802 | 0.993 | **0.782** | 0.853 | 0.803 | 0.827 |
| CHEMBL238_Ki | 0.682 | 0.709 | 0.723 | 0.748 | **0.681** | 0.748 |
| CHEMBL239_EC50 | **0.829** | 0.948 | 0.899 | 0.996 | 0.938 | 0.940 |
| CHEMBL244_Ki | 0.803 | 1.138 | **0.762** | 0.888 | 0.853 | 0.881 |
| CHEMBL262_Ki | 0.864 | 1.143 | 0.921 | 0.928 | **0.736** | 0.799 |
| CHEMBL264_Ki | 0.695 | 0.886 | **0.689** | 0.768 | 0.721 | 0.714 |
| CHEMBL2835_Ki | **0.755** | 0.927 | 0.905 | 0.959 | 1.029 | 0.921 |
| CHEMBL287_Ki | 0.742 | 0.849 | 0.824 | 0.886 | **0.699** | 0.713 |
| CHEMBL2971_Ki | **0.685** | 0.857 | 0.831 | 0.899 | 0.846 | 0.930 |
| CHEMBL3979_EC50 | 0.707 | 0.859 | **0.702** | 0.777 | 0.703 | 0.783 |
| CHEMBL4005_Ki | **0.751** | 0.875 | 0.804 | 0.768 | 0.791 | 0.810 |
| CHEMBL4203_Ki | 1.212 | 1.215 | 1.228 | 1.173 | 1.204 | **1.110** |
| CHEMBL4792_Ki | **0.647** | 0.729 | 0.687 | 0.798 | 0.674 | 0.734 |

Table 16: Comparison of MAE for GIN baseline vs. MALT-GIN on the Lo-Hi Benchmarks. Performance gains are shown for MALT. Best results are in **bold**.

| Split | Dataset | GIN (Baseline) | MALT-GIN (Ours) | Gain (%) |
|---|---|---|---|---|
| HI (Realistic OOD) | BACE | **0.8476 ± 0.0343** | 1.1882 ± 0.0488 | -40.2% |
| | ESOL | 0.4088 ± 0.0054 | **0.3979 ± 0.0087** | 2.7% |
| | FreeSolv | 0.3571 ± 0.0268 | **0.3010 ± 0.0111** | 15.7% |
| | Lipo | **0.5211 ± 0.0047** | 0.5758 ± 0.0145 | -10.5% |
| LO (Realistic OOD) | BACE | 0.7158 ± 0.0097 | **0.6791 ± 0.0181** | 5.1% |
| | ESOL | 0.3423 ± 0.0099 | **0.3267 ± 0.0138** | 4.6% |
| | FreeSolv | 0.4069 ± 0.0128 | **0.2393 ± 0.0118** | 41.2% |
| | Lipo | 0.5012 ± 0.0057 | **0.4598 ± 0.0067** | 8.3% |

## M  COMPUTATIONAL OVERHEAD ANALYSIS

To address the scalability and practicality of our framework, we analyzed the computational overhead in three key areas: training, inference, and memory bank construction. Our findings show that the framework scales favorably and can be optimized for large-scale applications.

### M.1  TRAINING OVERHEAD: ANCHOR SELECTION & MEMORY UPDATES

To assess training overhead, we compared our transductive MALT framework against a standard inductive GNN on two datasets of vastly different sizes: **BACE** (1,513 molecules) and **QM9** (133,885 molecules). The primary additional costs of our framework are **Memory Update** and **Anchor Selection**. As shown in Tables 17, the total overhead per epoch was +52% on the small BACE dataset and +43% on the large QM9 dataset, indicating favorable scaling as the dataset size grows. However, as the dataset size increases, we can adjust the framework to update the memory bank every N epochs for efficient learning. For all the experiments we have conducted, we updated the memory bank every epoch.

Table 17: Per-Epoch Training Time Breakdown. Values in each cell are shown for the BACE dataset → QM9 dataset.

| Training Component | Inductive Model (s) | MALT Framework (s) | Overhead (s) |
|---|---|---|---|
| **Forward Pass (Total)** | $0.0693 \rightarrow 14.9002$ | $0.0800 \rightarrow 17.7365$ | $+0.0107 \rightarrow +2.8364$ |
| *Query Embedding Extraction* | $0.0000 \rightarrow 0.0000$ | $0.0633 \rightarrow 9.0935$ | $+0.0633 \rightarrow +9.0935$ |
| *Anchor Selection* | $0.0000 \rightarrow 0.0000$ | $0.0052 \rightarrow 7.0040$ | $+0.0052 \rightarrow +7.0040$ |
| *Prediction Head* | $0.0000 \rightarrow 0.0000$ | $0.0115 \rightarrow 1.6390$ | $+0.0115 \rightarrow +1.6390$ |
| **Backward Pass** | $0.1241 \rightarrow 8.3738$ | $0.1340 \rightarrow 8.1957$ | $+0.0099 \rightarrow -0.1781$ |
| **Memory Update** | $0.0000 \rightarrow 0.0000$ | $0.0993 \rightarrow 12.8468$ | $+0.0993 \rightarrow +12.8468$ |
| **Data Loading** | $0.0065 \rightarrow 5.6589$ | $0.0042 \rightarrow 4.2272$ | $-0.0023 \rightarrow -1.4316$ |
| **Other Overhead** | $0.0367 \rightarrow 4.1498$ | $0.0418 \rightarrow 4.2201$ | $+0.0051 \rightarrow +0.0703$ |
| **TOTAL EPOCH TIME** | $\mathbf{0.2366 \rightarrow 33.0827}$ | $\mathbf{0.3592 \rightarrow 47.2264}$ | $\mathbf{+0.1226 \rightarrow +14.1437}$ |

### M.2  INFERENCE OVERHEAD: BRUTE-FORCE VS. FAISS OPTIMIZATION

To quantify inference overhead, we benchmarked the **Anchor Selection** step (nearest-neighbor search). Our brute-force PyTorch implementation was compared against **Faiss** (35) library over memory banks ranging from 10k to 10M vectors for different embedding dimensions. As shown in Table 18, for smaller memory banks ($\leq$ 100k vectors), our simple implementation is often faster. However, for larger banks and higher dimensions, Faiss provides a significant speedup, demonstrating a clear path to optimization for production-level applications.

Table 18: Faiss vs. PyTorch Nearest-Neighbor Search. Values in each cell are shown for embedding dimensions $D = 256 \rightarrow D = 128 \rightarrow D = 64$.

| Bank Size | PyTorch Init (ms) | Faiss Init (ms) | PyTorch Search (ms) | Faiss Search (ms) | Speedup |
|---|---|---|---|---|---|
| 10k | $1.84 \rightarrow 1.89 \rightarrow 1.70$ | $91.5 \rightarrow 160 \rightarrow 93.1$ | $0.22 \rightarrow 0.16 \rightarrow 0.15$ | $0.32 \rightarrow 0.31 \rightarrow 0.36$ | $0.68\times \rightarrow 0.50\times \rightarrow 0.41\times$ |
| 100k | $16.9 \rightarrow 20.2 \rightarrow 12.4$ | $162 \rightarrow 228 \rightarrow 116.5$ | $1.45 \rightarrow 1.00 \rightarrow 0.62$ | $1.47 \rightarrow 1.76 \rightarrow 2.26$ | $0.99\times \rightarrow 0.57\times \rightarrow 0.27\times$ |
| 500k | $128.7 \rightarrow 86.3 \rightarrow 94.5$ | $394 \rightarrow 306 \rightarrow 212$ | $7.24 \rightarrow 5.09 \rightarrow 3.09$ | $6.57 \rightarrow 7.79 \rightarrow 10.28$ | $\mathbf{1.10}\times \rightarrow 0.65\times \rightarrow 0.30\times$ |
| 1M | $175.1 \rightarrow 148.2 \rightarrow 129.6$ | $672 \rightarrow 506 \rightarrow 382$ | $14.48 \rightarrow 10.20 \rightarrow 6.14$ | $12.94 \rightarrow 15.44 \rightarrow 20.39$ | $\mathbf{1.12}\times \rightarrow 0.66\times \rightarrow 0.30\times$ |
| 10M | $2706.7 \rightarrow 2240.5 \rightarrow 1308.4$ | $6317 \rightarrow 4652 \rightarrow 2653$ | $134.70 \rightarrow 102.20 \rightarrow 61.44$ | $127.70 \rightarrow 152.00 \rightarrow 202.10$ | $\mathbf{1.05}\times \rightarrow 0.67\times \rightarrow 0.30\times$ |

### M.3  MEMORY BANK CONSTRUCTION: BUILD TIME & MEMORY FOOTPRINT

We also evaluated the one-time cost of building the memory bank on datasets ranging from $10^3$ to $10^6$ molecules. This procedure is performed once with gradients disabled to extract and store each molecule's embedding. As shown in Table 19, the process scales linearly with the number of molecules, with embedding extraction being the main bottleneck ( 71% of the total time).

Table 19: Memory Bank Construction Time and Size Scaling.

| Dataset Size | Total Time (s) | Data Loading (s) | Embedding Extraction (s) | Concatenation (s) | Memory Size (MB) |
|---|---|---|---|---|---|
| 1,000 | 0.274 | 0.004 | 0.194 | 0.028 | 0.98 |
| 5,000 | 0.771 | 0.006 | 0.540 | 0.014 | 4.88 |
| 10,000 | 1.594 | 0.007 | 1.106 | 0.018 | 9.77 |
| 50,000 | 7.709 | 0.044 | 5.517 | 0.148 | 48.83 |
| 100,000 | 15.538 | 0.096 | 10.872 | 0.321 | 97.66 |
| 500,000 | 75.837 | 0.396 | 53.686 | 1.782 | 488.28 |
| 1,000,000 | 151.713 | 0.790 | 108.333 | 3.147 | 976.56 |

