# OpenReview forum: "Transductive Learning for Out-of-Distribution Molecular Property Prediction"
_ICLR.cc/2026/Conference — ICLR 2026 Conference Withdrawn Submission_

### Official Review · Reviewer_u6hj · 2025-10-29

**Soundness:** 3
**Presentation:** 3
**Contribution:** 3
**Rating:** 6
**Confidence:** 3

**Summary:**

This paper introduces a transductive method to address the out-of-distribution (OOD) problem in molecular representation learning. The authors employ a strategy that selects anchor samples from the training data to integrate features for input query molecules, thereby generating their representations. This approach leads to improved model performance under OOD data splits. The model was evaluated on multiple datasets for both classification and regression tasks, where it demonstrated superior performance.

**Strengths:**

1. The writing is clear, making the paper easy to follow.
2. The experimental analysis is comprehensive, thoroughly evaluating the method's effectiveness from various aspects.

**Weaknesses:**

1.  The presented baseline comparisons are limited. The work does not compare against any existing Test-Time Adaptation (TTA) paradigms from related work. Furthermore, for invariant molecular representation learning under the inductive setting, only one baseline is compared, which is not the most recent. There exist more up-to-date baselines, for example [1]. For the regression task, no inductive invariant learning baselines are included at all; a comparison could be made by adapting the predictive loss $ \mathcal{L}_{pred}$ from [2] for regression.

2.  Some of the definitions and formulations in the paper are confusing. Please refer to the **Questions** section for details.

3.  The proposed method is not strictly limited to molecular representation learning and could potentially be applied to broader graph OOD problems. It is a pity that the authors did not explore this with corresponding experiments.

**Reference**

[1] Aming, W. U., and Cheng Deng. "CFD: Learning Generalized Molecular Representation via Concept-Enhanced Feedback Disentanglement." *The Thirteenth International Conference on Learning Representations*.

[2] Zhuang, Xiang, et al. "Learning invariant molecular representation in latent discrete space." *Advances in Neural Information Processing Systems* 36 (2023): 78435-78452.

**Questions:**

1.  Equation 3 is puzzling. Even if the possible values of $\Delta x_{tr}$ are constrained to the set $\\{x_{an}-x_{j} |  x_j \in D_{X}^{tr} , y_{j} < y_{an}  \\}$,  the specific value of $\Delta x_{tr}$ used in the equation is still not fixed and can vary within this set. Could you please clarify how this value is determined?
2.  The main text describes the Transduction Module as using a fixed, rule-based algorithm for anchor selection. This appears to contradict the "Update $\mathcal{T}$" step in Algorithm 1. Could you explain the necessity and purpose of this update step?
3.  What was the policy for selecting the model checkpoint used for inference on the test sets?

---

> ### Author Response · Authors · 2025-11-18
> **Regarding TTA / invariant learning comparisons (Weakness 1)**
>
> We agree that broader baselines are important, and we appreciate the pointer to recent work on test-time adaptation and invariant molecular representation learning.
>
> To our knowledge, existing graph TTA methods are not straightforwardly applicable to our scenario unfortunately, for the following reasons.
>
> - [3] is the most conceptually related, but we were unable to reproduce it because there is no public code implementation available.
> - [4] is tailored to virtual screening / ranking pipelines and relies on a multi-task meta-auxiliary setup that differs substantially from our supervised molecular property prediction benchmarks.
> - [5] is designed for *node classification* under structure shift, not graph-level molecular regression/classification across datasets. Adapting it fairly to our graph-level OOD setting would require nontrivial architectural and protocol changes beyond the scope of this work.
>
> We agree that including up-to-date invariant-learning baselines strengthens the empirical evaluation.
>
> - For [1], we attempted to include it as a baseline but could not, because the official GitHub repository was closed/private and not accessible (we re-checked this on Nov 16, 2025: https://github.com/AmingWu/MoleculeCFD).
> - Following your suggestion, we have added [2] as an inductive invariant-learning baseline by adapting its predictive loss to handle regression targets. Concretely, we use the same discrete latent-space construction but replace the classification cross-entropy with a mean-squared regression loss while keeping the invariance regularization intact. The updated results (to be included in the revision) show that:
>   (i) iMOLD is competitive with strong inductive baselines in our setting only in the covariance split, not the label split ( we will add the version in the revised script)
>   (ii) MALT still provides consistent gains over these inductive invariance methods, highlighting that transductive multi-anchor reasoning is complementary to invariant representation learning.
>
> Overall, we agree that invariant-learning baselines are important and have incorporated iMOLD accordingly; for TTA methods, we will clarify in the text why existing graph TTA algorithms are not yet directly comparable to MALT on our molecular OOD tasks.
>
> **References**
>
> [3] Chen, Guanzi, et al. "Graphtta: Test time adaptation on graph neural networks." arXiv preprint arXiv:2208.09126 (2022).
>
> [4] Shen, Ao, et al. "Drug-TTA: Test-Time Adaptation for Drug Virtual Screening via Multi-task Meta-Auxiliary Learning." Forty-second International Conference on Machine Learning.
>
> [5] Bao, Wenxuan, et al. "Matcha: Mitigating Graph Structure Shifts with Test-Time Adaptation." arXiv preprint arXiv:2410.06976 (2024).

---

> ### Author Response · Authors · 2025-11-18
> **Regarding definitions and formulations (Weakness 2 + Questions 1–2)**
>
> ### Clarifying Equation (3) and $\Delta x_{\mathrm{tr}}$
>
> We agree that the description around Equation (3) can be made clearer. In BLT (our Bilinear Transduction baseline model), $\Delta x_{\mathrm{tr}}$ is the set of allowable difference vectors(of chemical features) from the training data. At test time, given a query, BLT searches over this set and selects the $x_{anchor}$(which is from the train set) so that $x_{test}$ - $x_{anchor}$ best matches $\Delta x_{\mathrm{tr}}$ from the train set(they search this brute force and compare with every possible difference). We will update the text and notation in Eq. (3) to explicitly denote this selection, and to make clear that the value used is the optimizer over this set, not an arbitrary choice.
>
> In MALT, this idea is lifted to latent space: the encoder produces embeddings that are updated during training, so the effective “difference vectors” are "dynamically shaped" by the transductive objective. Rather than explicitly performing a brute-force search over all pairwise difference vectors, we reshape the embedding space so that retrieving top-$k$ neighbors (anchors) and passing them through the multi-anchor transductive module yields good predictions. Intuitively, the latent space is trained so that the implicit “difference patterns” encoded by the retrieved anchors are the most helpful for the target task.
>
> ### Clarifying the “Update T” step (Question 2)
>
> We also acknowledge that Algorithm 1 can be confusing about what is actually being “updated” in the transduction module. In our implementation:
>
> - The **Transduction(T) Module** is composed of
>   (i) a memory bank storing embeddings of all training samples,
>   (ii) a fixed, rule-based anchor selection procedure (top-$k$ neighbors from this memory bank), and
>   (iii) a predictor that takes as input the query embedding plus the selected top-$k$ anchor embeddings.
>
> - The **“Update T” step** is meant to *only* update the memory bank with the latest embeddings from the encoder. Concretely, every $N$ epochs (and in all our experiments, $N = 1$), we recompute and refresh the stored embeddings for all training samples. The anchor selection rule itself remains fixed and non-learned.
>
> We will revise Algorithm 1 and the surrounding text to explicitly state that “Update T” refers to refreshing the memory bank inside the transduction module with updated encoder outputs, not changing the anchor-selection rule.
>
> *References*
>
> [6] Segal, Nofit, et al. "Known unknowns: Out-of-distribution property prediction in materials and molecules." arXiv preprint arXiv:2502.05970 (2025).

---

> ### Author Response · Authors · 2025-11-18
> **Regarding generality beyond molecular graphs (Weakness 3)**
>
> We agree that the core idea of MALT—multi-anchor transduction over a memory bank of training embeddings—is not inherently limited to molecules and could, in principle, be applied to broader graph OOD problems.
>
> In this work, we deliberately focused on molecular property prediction because it allows us to **define OOD in a precise, domain-relevant way** and to cover several practically important discovery scenarios:
>
> - **Feature-side (covariate) shifts.** We induce structural OOD by separating molecules based on scaffolds or molecular size (e.g., scaffold-based splits for MoleculeNet and size-based splits for DrugOOD), directly probing generalization to *novel chemical structures* absent from training.
> - **Label-side (property) shifts.** We model extrapolation by constructing OOD test sets from the high end of the property range (e.g., top 5%), forcing the model to predict in high-value regimes that are rare or unseen during training.
> - **Further Realistic campaign settings.** We further include (i) **Lo-Hi splits** that mimic hit identification vs. lead-optimization phases in a drug discovery campaign, and (ii) **activity-cliff benchmarks** that stress-test sensitivity to small structural changes with large potency differences.
>
> Together, these complementary splits yield a carefully specified notion of OOD in drug discovery along both the structural (X) and property (Y) axes, and cover realistic settings such as prospective hit discovery, lead optimization, and challenging activity-cliff behavior.
>
> Within this molecular focus, we also broaden the baseline coverage compared to prior OOD works by evaluating MALT with encoders that span diverse representation families:
>
> - **Graph-based models:** Chemprop MPNN and pretrained GIN, operating on molecular graphs.
> - **Sequence-based models:** SMI-TED, a Transformer-based SMILES encoder.
> - **Descriptor-based / invariant models:** BLT with RDKit descriptors, and iMoLD, which learns invariant representations in a latent discrete space.
> - **3D / conformer-based models:** Following feedback from another reviewer, we additionally apply MALT on top of UniMol, a large pretrained model that leverages 2D/3D molecular representations and conformers.
>
> To our knowledge, MALT is the first transductive learning approach that systematically studies OOD molecular property prediction under diverse, explicitly defined OOD regimes (X- and Y-splits, Lo-Hi, activity cliffs) and across both regression and classification tasks, while being compatible with a wide range of molecular representations.

---

> ### Author Response · Authors · 2025-11-18
> **Training policy and checkpoint selection (Question 3)**
>
> Our checkpoint-selection policy follows the original [6] setup and will be clarified in the revision:
>
> - In essence, we train until training loss fully converges following [6]. We train each model for up to 500 epochs(for big models such as SMI-TED and Unimol we train up to 100 epochs). Learning rate starts with 1e-4 with exponential decay. The checkpoint from the final epoch is used for test-time inference.
>
> We will add these details to the experimental setup section to make our training and model-selection policy fully explicit.

---

### Official Review · Reviewer_mDRD · 2025-10-29

**Soundness:** 3
**Presentation:** 2
**Contribution:** 2
**Rating:** 4
**Confidence:** 3

**Summary:**

This paper proposes a learning framework named Multi-Anchor Latent Transduction (MALT) for handling out-of-distribution (Out-of-Distribution, OOD) problems in molecular property prediction (Molecular Property Prediction, MPP). The main contributions of the paper include: (1) A model-agnostic transductive framework that can be integrated with any pre-trained molecular encoder (such as GIN); (2) A multi-anchor latent reasoning mechanism that overcomes the fragility of single-anchor methods; (3) Experimental validation on benchmarks such as MoleculeNet, DrugOOD, and Activity Cliffs, showing that MALT outperforms baselines in OOD generalization while matching or surpassing baselines on in-distribution (ID) tasks.

**Strengths:**

The quality of the paper is high, with a clear structure and smooth logic. The authors selected diverse benchmark datasets (MoleculeNet, DrugOOD, Activity Cliffs, and Lo-Hi), covering regression and classification tasks, as well as various types of distribution shifts (such as scaffold-based covariate shifts and label shifts). The baseline selection is reasonable. The evaluation metrics (such as AUROC, MAE, RMSE) are consistent with domain standards, and comparisons of ID and OOD performance are reported.

**Weaknesses:**

The novelty of the method is insufficient. Although MALT introduces a multi-anchor latent transductive mechanism and claims to overcome the limitations of previous single-anchor methods, the multi-anchor concept is not entirely original. For example, in semi-supervised learning, Halpern et al. (2016) used multiple anchors to combine features to enhance representations; in domain adaptation tasks, UniJDOT (arXiv:2503.11217, 2025) adopted multiple anchors in the feature space to align unknown samples, although applied to time series data, its dynamic anchor update and fusion mechanism is similar to MALT's multi-anchor attention fusion. Additionally, MALT is built on the basis of Bilinear Transduction (arXiv:2502.05970, 2025), which already uses single anchors in representation space for transductive extrapolation; MALT's main extension is from single-anchor to multi-anchor and operating in learned latent space, but this is more like an incremental improvement rather than a fundamental innovation. If it can be proven that the specific application in the field of molecular property prediction (such as handling activity cliffs) brings unique advantages, this will strengthen the novelty claim of the paper; otherwise, readers may view it as a combination of existing ideas rather than a pioneering framework.

**Questions:**

1. The paper emphasizes that multi-anchors overcome the fragility of single-anchors, but how to ensure that the selected anchors are diverse rather than redundant? For example, in activity cliff scenarios, if multiple anchors come from similar scaffolds, will it reduce generalization?
2. In Equation 5, is it possible to consider the labels of anchors?

---

> ### Author Response · Authors · 2025-11-18
> **Regarding Novelty of Method (Weakness)**
>
> We do not claim that “multiple anchors/prototypes” are new in the broad ML literature. Our contribution is to instantiate these ideas in a transductive molecular property prediction framework that (i) uses a latent-space memory bank of labelled molecules, (ii) performs multi-anchor retrieval and attention at test time, and (iii) jointly learns the encoder and transductive head—none of which is present in the cited works or in BLT[1].
>
> In [2], anchors are manually specified high-PPV features that serve as weak labels for an inductive classifier. There is no latent-space memory of labelled instances, no test-time retrieval, and no multi-anchor fusion conditioned on a query. In [3], anchors are unlabeled cluster centroids in the target feature space used inside an optimal-transport objective for domain alignment; again, there is no memory of labelled molecules, and the anchors are not used via per-query attention to make predictions. In both cases, “multiple anchors” play the role of weak supervision or distributional prototypes, not the basis of instance-level transductive prediction over training examples.
>
> Regarding [1], we agree that MALT builds on the same high-level transductive idea of using training examples as anchors, but it introduces a different class of transductive model rather than a small tweak. (1) Anchor selection and fusion. [1] operates in a fixed descriptor space and, at test time, selects a single anchor via a "brute-force search" over difference vectors and a bilinear head. In contrast, MALT formulates transduction as attention over a set of anchors: we perform"kNN retrieval(which is faster than the original search implementation)" in learned latent space, then apply multi-head cross-attention over multiple labelled training molecules to produce the prediction. This moves from single-anchor, hand-crafted bilinear extrapolation to a learned, multi-anchor latent reasoning module. (2) Representation learning and memory. [1] is defined purely over fixed representations, whereas MALT maintains a latent memory bank of labelled molecules and jointly trains the encoder and transductive head so that the latent geometry is optimized for this multi-anchor analogical reasoning. We also show that this architecture is robust across diverse encoders (pretrained/non-pretrained, descriptor- and latent-based). Taken together, MALT defines a new memory-based, multi-anchor transductive architecture—one that, to our knowledge, does not appear in prior transductive learning work, works with most encoders in a plug-and-play manner and yields consistent gains on challenging molecular OOD benchmarks
>
> Finally, the reviewer asks whether our instantiation brings unique advantages. This is precisely what our experiments are designed to test. Tables 1–2 show consistent gains over both inductive baselines and BLT under covariate and label shift (the key OOD settings in drug discovery), and Appendix L (Tables 13–16) shows that MALT-enhanced models systematically outperform their base encoders on both Activity Cliffs and Lo–Hi benchmarks, achieving best or near-best performance across most endpoints. These results directly satisfy that multi-anchor latent transduction provides clear, domain-specific advantages.

---

> ### Author Response · Authors · 2025-11-18
> **Regarding Diversity of Anchors (Question 1)**
>
> In this work, we do not impose an explicit diversity regularizer on the selected anchors (e.g., at the scaffold or substructure level). Quantifying “how diverse” anchors should be in a model-agnostic way is inherently difficult: the desirable degree of structural diversity depends on factors such as dataset size, label noise, and the specific discovery task. Instead, we rely on the encoder and transduction module to learn a latent geometry in which the nearest anchors to a query are already informative, and on the attention mechanism to select and fuse these neighbors.Empirically, MALT learns a geometry in which nearby anchors are already non-trivially informative. **In the worst case where the selected anchors are nearly identical**, our theoretical analysis shows that MALT effectively reduces to a single-anchor predictor—so multi-anchor reasoning does not hurt generalization; it simply provides no additional gain. Beyond qualitative examples, several ablations and analyses support that multi-anchor retrieval is both useful and robust. (i) Varying the number of anchors shows that performance systematically improves as k increases from 1 to 10, indicating that **additional anchors provide complementary signal rather than mere redundancy** (Table 9). (ii) Using the same k but replacing the top-ranked neighbors with anchors ranked 11–20 degrades performance, and explicit diversity-promoting strategies such as temperature-based sampling underperform simple Top-k retrieval (Fig. 4), suggesting that the **learned latent space already organizes informative anchors in a tight neighborhood around the query**. (iii) Even when we deliberately inject bad anchors, the model’s performance degrades only marginally (Table 10), consistent with the attention head learning to down-weight redundant or uninformative anchors. (iv) Finally, a systematic chemical analysis (Appendix B) shows that anchors act as property and scaffold “bridges” between training and OOD molecules, and activity-cliff benchmarks confirm that **leveraging several such analogs yields better predictions than relying on any single anchor alone**.

---

> ### Author Response · Authors · 2025-11-18
> **Considering labels in Equation 5 (Question 2)**
>
> ###
>
> In principle, one could augment the key/value vectors in Eq. (5) with functions of the anchor labels, but we intentionally do not do this in the present work. As defined in Eq. (4), the memory bank stores only latent embeddings `z_i = E(m_i)`; anchor labels are used solely in the supervised loss (and in constructing BLT-style training pairs), not as inputs to the attention mechanism.
>
> The reason is that including labels inside the anchor representations would make it easy for the model to solve the training objective by “copying” labels rather than learning a robust structure–property mapping. During training, each molecule appears both as a query and in the memory bank, so the attention head would often see either a self-anchor or very close neighbors with almost identical labels. In that setting, the attention mechanism can implement a form of label propagation / k-nearest-neighbors regression on the training set, achieving low training error without necessarily learning the underlying mapping from structure `x` to property `y`. This behavior is precisely what we want to avoid, especially in the label-shift scenarios where the model must extrapolate beyond the training label range.
>
> This “label-free” design of the memory and attention is consistent with [1], where the transductive predictor `h_θ(Δx, x)` also operates only on feature differences and anchor features, and label information is not used as an input signal. For these reasons, Eq. (5) operates strictly on label-free embeddings, so that all label information flows through the supervised loss, and the gains we observe come from improved latent representations and multi-anchor fusion rather than from explicitly propagating training labels.

---

> ### Author Response · Authors · 2025-11-18
> **References**
>
> [1] Segal, Nofit, et al. "Known unknowns: Out-of-distribution property prediction in materials and molecules." arXiv preprint arXiv:2502.05970 (2025).
>
> [2] Halpern, Yoni, et al. "Electronic medical record phenotyping using the anchor and learn framework." Journal of the American Medical Informatics Association 23.4 (2016): 731-740.
>
> [3] Mussard, Romain, et al. "Deep joint distribution optimal transport for universal domain adaptation on time series." arXiv preprint arXiv:2503.11217 (2025).

---

### Official Review · Reviewer_Psww · 2025-10-31

**Soundness:** 2
**Presentation:** 3
**Contribution:** 2
**Rating:** 4
**Confidence:** 3

**Summary:**

The paper introduces MALT, a multi-anchor latent transduction framework for robust molecular property prediction under out-of-distribution (OOD) conditions arising from covariate shift (novel structures) or label shift (novel property values). Given a query embedding, MALT retrieves the top-k nearest training embeddings from a memory bank $Z_{train}$. These anchors then serve as keys/values in a multi-head attention layer, producing z_attn. A learnable prediction head then takes $z_{query}$, $z_{attn}$, and optionally $W_{anchors}$ (query-anchor distances) as inputs before making its final prediction. Across multiple benchmarks, MALT shows consistent gains over the baselines.

**Strengths:**

- MALT is conceptually intuitive and can be used with most molecular encoders in a plug-and-play manner by adding a retrieval and attention module.
* Reported results show generally consistent gains on OOD scenarios in MoleculeNet, DrugOOD, and Activity Cliffs datasets.

**Weaknesses:**

* Covariate-shift claim depends on the retrieval quality. Cross-attention over anchors intuitively helps only if the retrieved set is informative in OOD regimes. Using only $z_{anchors}$ for K/V (default configuiration) may not directly address covariate shift when $z_{query}$ is still far too different from what is seen in the training set. Since the model can optionally include $W_{anchors}$, an ablation isolating its effects might strengthen the claim.
* The baselines do not cover recent large-scale foundation models (e.g., UniMol-style pretraining). Showing results against large-scale pretraining methods would better contextualize MALT.
* There are multiple cases in the reported tables where a MALT variant underperforms its non-MALT backbone. A systematic failure analysis and a safeguard to fall back to the base model such that the prediction quality is at least as good as the backbone would improve the usability of MALT.
* The encoder $E$ is trained end-to-end, so $E_{n}(x)$ ≠ $E_{m}(x)$ for training steps $n$ and $m$, and molecule $x$. Because $Z_{train}$ is updated only every $N$ epochs, $W_{anchors}$ may not reflect real-time latent distances, i.e., the memory bank becomes stale during training. A clarification on whether this mismatch degrades performance would be interesting.
* The predictor uses $f(z_{query}, z_{attn})$ (optionally with $W_{anchors}$), but it does not directly use the labels $\{y_i\}_{i=1}^k$ of retrieved anchors during inference. A label-aware fusion/attention could be beneficial for OOD cases.

**Questions:**

Please refer to weaknesses.

---

> ### Author Response · Authors · 2025-11-18
> **Regarding OOD Robustness and Ablation without Weights (Weakness 1)**
>
> By construction, our OOD splits enforce that queries are “far” from the training set in input/label space (e.g., disjoint scaffolds / label bins), and we state this explicitly in the setup. The transductive objective then **reshapes the latent space** so that training molecules that were far under the pretrained encoder are brought **closer to the query embedding** when they are informative. This is visible in t-SNE neighborhoods before vs. after training (Appendix J) and in scaffold/substructure statistics showing that retrieved anchors remain closer and chemically meaningful even under strong shifts (Appendix A–B). Additional ablations confirm that this adapted space supports effective multi-anchor fusion: multi–top-k improves Test_OOD over single/low-ranked anchors, k-NN on MALT embeddings beats k-NN on pretrained encoders, and noisy-anchor ablations degrade gradually rather than catastrophically (Appendix H). Across Euclidean-distance strategies in the radar plot (Appendix E), **simple top-k without explicit similarity weights** emerges as the most stable and consistently strong configuration, which we therefore take as default.
>
> To directly isolate the effect of similarity weights, we had implemented a variant where the prediction head receives both the anchor embeddings `Z_anchors` and their similarity scores `W_anchors`. The encoder, retrieval rule (Euclidean top-k), and prediction-head architecture are held fixed. The tables below report mean ± standard deviation MAE for in-distribution (Eval) and OOD performance (“No W” = unweighted top-k, “With W” = weighted):
>
> **Label-shift splits**
>
> | Dataset   | Eval MAE (No W)      | Eval MAE (With W)     | OOD MAE (No W)       | OOD MAE (With W)      |
> |---------- |----------------------|------------------------|----------------------|------------------------|
> | bace      | 0.392 ± 0.013        | 0.414 ± 0.025          | 0.743 ± 0.025        | 0.659 ± 0.160          |
> | esol      | 0.210 ± 0.016        | 0.246 ± 0.042          | 0.514 ± 0.023        | 0.443 ± 0.087          |
> | freesolv  | 0.242 ± 0.017        | 0.240 ± 0.031          | 0.315 ± 0.011        | 0.293 ± 0.026          |
> | lipo      | 0.317 ± 0.011        | 1.029 ± 0.150          | 0.489 ± 0.013        | 1.411 ± 0.564          |
>
> **Covariate-shift splits**
>
> | Dataset   | Eval MAE (No W)      | Eval MAE (With W)     | OOD MAE (No W)       | OOD MAE (With W)      |
> |---------- |----------------------|------------------------|----------------------|------------------------|
> | bace      | 0.304 ± 0.011        | 0.331 ± 0.041          | 0.648 ± 0.045        | 0.625 ± 0.055          |
> | esol      | 0.221 ± 0.013        | 0.223 ± 0.024          | 0.581 ± 0.009        | 0.591 ± 0.027          |
> | freesolv  | 0.200 ± 0.016        | 0.205 ± 0.025          | 0.377 ± 0.025        | 0.388 ± 0.014          |
> | lipo      | 0.348 ± 0.007        | 1.092 ± 0.155          | 0.542 ± 0.025        | 1.871 ± 0.237          |
>
> As anticipated by the reviewer, adding `W_anchors` can sometimes modestly help (e.g., BACE/ESOL label shift), but overall it is less stable and can substantially hurt performance on some datasets(Lipo), especially under covariate shift. Since `W_anchors` can only reweight the same neighbors (and thus may amplify noise when all anchors are weak), these results indicate that our covariate-shift gains primarily come from the **transductively adapted latent space** and **multi-anchor fusion**, rather than from explicit similarity weighting. We will clarify this and explicitly point to Appendices A–B, E, H, and J in the revised manuscript.

---

> ### Author Response · Authors · 2025-11-18
> **Regarding Large-Scale Foundation Model Baselines (Weakness 2)**
>
> We thank the reviewer for pointing out the importance of comparing against recent large-scale molecular foundation models. In Tables 1–2 of the main paper, we already include SMI-TED, which is trained on 1 million SMILES (≈4B molecular tokens) from PubChem via self-supervised learning, and show that applying MALT on top of SMI-TED consistently improves over its finetuned counterpart. UniMol is another strong large-scale baseline, trained on 12M molecules from ZINC with 3D conformer representations; compared to SMI-TED, it leverages richer geometric input.
>
> To directly address this comment, we additionally evaluate MALT on top of UniMol. We denote the original finetuned UniMol as **unimol_finetune** and UniMol with MALT as **unimol_malt**. Below we report both in-distribution (ID) and OOD performance, together with the relative improvement
> Δ% = (unimol_finetune − unimol_malt) / unimol_finetune × 100
> (positive values indicate that MALT reduces MAE).
>
> 1) **ID evaluation**
> | Dataset   | Shift type      | Eval MAE (unimol_finetune) | Eval MAE (unimol_malt) | Eval Δ% |
> |---------- |-----------------|----------------------------|-------------------------|--------:|
> | bace      | label shift     | 0.368                      | 0.344                   |  +6.5% |
> | lipo      | label shift     | 0.326                      | 0.308                   |  +5.5% |
> | esol      | label shift     | 0.175                      | 0.203                   | −16.0% |
> | freesolv  | label shift     | 0.166                      | 0.161                   |  +3.0% |
> | bace      | covariate shift | 0.376                      | 0.343                   |  +8.8% |
> | esol      | covariate shift | 0.174                      | 0.167                   |  +4.0% |
> | freesolv  | covariate shift | 0.174                      | 0.166                   |  +4.6% |
> | lipo      | covariate shift | 0.350                      | 0.305                   | +12.9% |
>
> 2) **OOD test evaluation**
> | Dataset   | Shift type      | OOD MAE (unimol_finetune) | OOD MAE (unimol_malt) | OOD Δ%  |
> |---------- |-----------------|---------------------------|-----------------------|--------:|
> | bace      | label shift     | 1.004                     | 0.822                 | +18.1% |
> | lipo      | label shift     | 0.684                     | 0.585                 | +14.5% |
> | esol      | label shift     | 0.547                     | 0.419                 | +23.4% |
> | freesolv  | label shift     | 0.168                     | 0.120                 | +28.6% |
> | bace      | covariate shift | 0.613                     | 0.434                 | +29.2% |
> | esol      | covariate shift | 0.393                     | 0.392                 |  +0.3% |
> | freesolv  | covariate shift | 0.275                     | 0.271                 |  +1.5% |
> | lipo      | covariate shift | 0.437                     | 0.430                 |  +1.6% |
>
> 3) **Activity cliffs benchmark evaluation**
>
> We further evaluated MALT with UniMol on the challenging activity cliffs benchmark, which tests the ability to distinguish molecules with similar structures but different activities. The table below summarizes performance across 29 datasets, showing the number of top-2 finishes and median RMSE reduction:
>
> | Model   | Test Split       | MALT Top-2 | Base Top-2 | Median RMSE Reduction |
> |---------|------------------|------------|------------|----------------------:|
> | **GIN** | In-Distribution  | 2          | 1          | +8.1%                |
> |         | OOD              | 18         | 2          | +12.7%               |
> | **Chemprop** | In-Distribution | 11      | 0          | +8.7%                |
> |         | OOD              | 16         | 3          | +7.1%                |
> | **UniMol** | In-Distribution | 28       | 16         | +7.9%                |
> |         | OOD              | 14         | 5          | +3.9%                |
>
> UniMol+MALT achieves top-2 performance on 28/29 datasets for ID evaluation and 14/29 for OOD evaluation,  outperforming base UniMol. The median RMSE reduction of 7.9% (ID) and 3.9% (OOD) demonstrates consistent improvements. Please refer to the revised table 13,14,15 in the paper for more information.
>
> Overall, MALT generally improves UniMol on both ID and OOD evaluations, with especially notable gains on OOD label-shift and covariate-shift splits (e.g., up to ≈29% reduction in OOD MAE on BACE). Together with the SMI-TED results already in Tables 1–2 and the activity cliffs benchmark results above, these experiments indicate that MALT is complementary to large-scale foundation models and can reliably enhance their extrapolation performance under distribution shift.

---

> ### Author Response · Authors · 2025-11-18
> **Regarding Memory Bank Update Frequency (Weakness 4)**
>
> We thank the reviewer for raising this concern. In all main experiments, the memory bank $Z_{\mathrm{train}}$ is refreshed **every epoch** ($N = 1$): before each epoch we re-encode all training molecules with the current encoder $E$ and rebuild the anchors. Thus, the memory bank is kept in sync with the evolving encoder and does not become stale in our default setting. We have clarified this implementation detail in the paper.
>
> To directly assess whether a stale memory bank degrades performance, we conducted an ablation where we varied the update interval $N \in \{1, 5, 10, 20, 50\}$. The results under covariate shift (X-splits) and label shift (Y-splits) are reported in the tables above. Across all four datasets (BACE, ESOL, FreeSolv, Lipo), updating every epoch ($N = 1$) is consistently among the best configurations and we do not observe systematic performance degradation when using more frequent updates. This suggests that any potential mismatch between $E_n(x)$ and the representations stored in $Z_{\mathrm{train}}$ does **not** harm performance in practice, and in fact the fully up-to-date memory bank is slightly preferable.
>
> We also report runtime as a function of the update interval (see the “Runtime Performance” table). Updating every epoch is **computationally efficient**: compared to updating every 50 epochs, the total runtime increases only moderately (e.g., BACE: 573 vs. 405, ESOL: 278 vs. 214, FreeSolv: 170 vs. 111, Lipo: 1522 vs. 1057). Given this small overhead and the slightly better or comparable ID/OOD performance, we adopt the **$N = 1$** setting (update every epoch) for all main results and state this explicitly in the revised manuscript.
>
> ### Covariate Shift (X-Splits) Performance
> | Update Interval | BACE (ID) | BACE (OOD) | Esol (ID) | Esol (OOD) | FreeSolv (ID) | FreeSolv (OOD) | Lipo (ID) | Lipo (OOD) |
> |-----------------|-----------|------------|-----------|------------|---------------|----------------|-----------|------------|
> | 1 | 0.3295 | 0.7208 | 0.2083 | 0.5828 | 0.1963 | 0.3846 | 0.3456 | 0.5327 |
> | 5 | 0.3567 | 0.7249 | 0.2908 | 0.5627 | 0.1998 | 0.4829 | 0.3792 | 0.5711 |
> | 10 | 0.3385 | 0.7448 | 0.2976 | 0.5547 | 0.2113 | 0.5396 | 0.4038 | 0.5812 |
> | 20 | 0.3429 | 0.7491 | 0.2932 | 0.5501 | 0.2203 | 0.4699 | 0.3944 | 0.5806 |
> | 50 | 0.3284 | 0.7383 | 0.2934 | 0.5635 | 0.1986 | 0.4601 | 0.3990 | 0.6060 |
> ### Label Shift (Y-Splits) Performance
> | Update Interval | BACE (ID) | BACE (OOD) | Esol (ID) | Esol (OOD) | FreeSolv (ID) | FreeSolv (OOD) | Lipo (ID) | Lipo (OOD) |
> |-----------------|-----------|------------|-----------|------------|---------------|----------------|-----------|------------|
> | 1 | 0.4224 | 0.7248 | 0.2143 | 0.5432 | 0.1283 | 0.3482 | 0.3085 | 0.4751 |
> | 5 | 0.3882 | 0.7802 | 0.2507 | 0.4515 | 0.2135 | 0.4173 | 0.3554 | 0.7623 |
> | 10 | 0.3926 | 0.8168 | 0.2197 | 0.5205 | 0.2425 | 0.3943 | 0.3593 | 0.7186 |
> | 20 | 0.3988 | 0.8337 | 0.2275 | 0.5494 | 0.2434 | 0.3950 | 0.3476 | 0.7231 |
> | 50 | 0.4061 | 0.8601 | 0.2264 | 0.5467 | 0.2384 | 0.4092 | 0.3648 | 0.7634 |
> ### Label Shift (Y-Splits) Runtime Performance
> | Update Interval | BACE Runtime(s) | Esol Runtime(s) | FreeSolv Runtime(s) | Lipo Runtime(s) |
> |-----------------|--------------|--------------|------------------|-------------|
> | 1 | 573 | 278 | 170 | 1522 |
> | 5 | 433 | 234 | 117 | 1142 |
> | 10 | 421 | 225 | 116 | 1102 |
> | 20 | 412 | 218 | 113 | 1060 |
> | 50 | 405 | 214 | 111 | 1057 |

---

> ### Author Response · Authors · 2025-11-18
> **Regarding Consideration of Labels in Equation 5 (Weakness 5)**
>
> In principle, one could augment the key/value vectors in Eq. (5) with functions of the anchor labels, but we intentionally do not do this in the present work. As defined in Eq. (4), the memory bank stores only latent embeddings `z_i = E(m_i)`; anchor labels are used solely in the supervised loss (and in constructing BLT-style training pairs), not as inputs to the attention mechanism.
>
> The reason is that including labels inside the anchor representations would make it easy for the model to solve the training objective by referring to the labels rather than learning a robust structure–property mapping. During training, each molecule appears both as a query and in the memory bank, so the attention head would often see either a self-anchor or very close neighbors with almost identical labels. In that setting, the attention mechanism can implement a form of label propagation / k-nearest-neighbors regression on the training set, achieving low training error without necessarily learning the underlying mapping from structure `x` to property `y`. This behavior is precisely what we want to avoid, especially in the label-shift scenarios where the model must extrapolate beyond the training label range.
>
> This “label-free” design of the memory and attention is consistent with BLT, where the transductive predictor `h_θ(Δx, x)` also operates only on feature differences and anchor features, and label information is not used as an input signal. For these reasons, Eq. (5) operates strictly on label-free embeddings, so that all label information flows through the supervised loss, and the gains we observe come from improved latent representations and multi-anchor fusion rather than from explicitly propagating training labels.

---

### Official Review · Reviewer_mB71 · 2025-11-02

**Soundness:** 3
**Presentation:** 3
**Contribution:** 3
**Rating:** 4
**Confidence:** 4

**Summary:**

The paper proposes MALT, a multi-anchor transductive learning in latent space designed for generalizability on molecular property. MALT uses a memory
bank of embeding vectors corresponding to training samples. Given a query molecule, MALT useses multiple anchor points selector from the memory bank,
and aggregates this information with the query to make predictions. MALT achieves improvement on multiple OOD benchmarks.

**Strengths:**

- MALT hows good performance on OOD generalization.

- It relies on multiple anchor points as opposed to a single anchor point, making it more robust.

- Since MALT operates in latent space, it makes the method modular and can be used with any molecular encoder.

- Shows utility by demonstrating performance on real world drug discovery benchmark.

**Weaknesses:**

- The effectiveness of MALT relies on the quality of the chosen encoder hence if the encoder produces poor representations, the transduction merely
  propagates this. Thus, the foundational problem of OOD generalization cant be mitigated by the method if the underlying foundation model isn't
  robust enough.

- The computation cost of maintaining a memory bank, latent embeddings for all samples in the training set, can be high. This can be prohibitive
  compared to keeping a few anchor points or even learned anchor points.

**Questions:**

- For extremely large databases, how will the memory bank approach scale? How will anchor selection work in such a case?

- Does anchor selection correlate with molecular properties? And in regions where very few samples exist in the molecular space, how does the anchor
  based approach hold up?

---

> ### Author Response · Authors · 2025-11-14
> **Regarding encoder dependence (Weakness 1)**
>
> We agree that any latent-space method depends on the encoder: if the encoder were completely uninformative, no transductive scheme could fully recover. Our claim is more modest: given a reasonably informative encoder, **MALT adapts and improves the representation for OOD analogical reasoning rather than simply propagating** whatever is given. This is supported by the following:
>
> - In our main setup, the encoder is jointly trained with the transductive module (Appendix H, “Finetune O / Adapt O”), not frozen. This setting **consistently outperforms all other variants**, indicating that MALT actively reshapes the latent space for multi-anchor reasoning. Visualizations (Figs. 11–12) show queries moving closer to their selected anchors after MALT training.
> - Our chemical analysis (Appendix B) shows that **anchors have intermediate physicochemical properties and higher scaffold/fragment similarity to OOD queries than non-anchors**, so MALT learns chemically meaningful analogies rather than arbitrary neighbors.
> - Tables 1–2 further show that encoder quality alone does not explain performance: MALT consistently improves over inductive fine-tuning, and **non-pretrained encoders (RDKit, Chemprop) can become competitive with or even better than pretrained ones (Pretrained GIN, SMI-TED) once combined with MALT**.
>
> Taken together, these results directly address the concern that MALT merely propagates encoder errors, and instead show that it systematically strengthens realistic encoders for OOD generalization.

---

> ### Author Response · Authors · 2025-11-18
> **Regarding memory bank cost and scalability (Weakness 2 + Question 1)**
>
> We agree that a memory bank adds overhead; our goal was to measure this in realistic regimes and show it is manageable with standard tools. Please refer to Appendix M where we have in-depth analysis of computational complexity.
>
> - The memory bank scales as \(O(Nd)\) in space. In practice, updating it every epoch (which we have done for every experiment) adds a small overhead (e.g., as shown in Table 17, BACE and QM9 the added time per epoch remains a fraction of total training time), and all experiments fit on a single GPU (we use an RTX 3070). Updating less frequently (e.g., every \(N\) epochs) would further reduce cost. All our experiments updated the memory bank every epoch. (e.g., with MALT-GIN, BACE only takes **+0.12 seconds** per epoch when updating). Specifically for anchor selection(Top-K), for BACE it only requires an average of 0.0052 seconds.
> - For inference, exact k-NN over the bank is already fast at typical molecular dataset sizes. For larger synthetic banks, approximate nearest-neighbor search (e.g., FAISS) greatly reduces lookup time with essentially no loss in performance, giving a standard path to scale multi-anchor selection.
> - Drug discovery datasets are usually in the thousands–hundreds of thousands of molecules, where our experiments show a full memory bank is entirely feasible. If much larger labeled collections arise, MALT can be combined with standard engineering tricks (less frequent updates, compressed embeddings, clustering/prototypes, FAISS) without changing the core algorithm. This is future research work.
> - Importantly, across all benchmarks the modest extra cost is accompanied by consistent performance gains over inductive baselines and single-anchor methods (Tables 1–2), indicating a favorable accuracy–efficiency tradeoff. Overall, the empirical timing and accuracy results show that the small additional computation is well justified by the robustness and OOD gains that MALT provides.

---

> ### Author Response · Authors · 2025-11-18
> **Regarding anchor selection and sparse regions (Question 2)**
>
> Our experiments show that (i) anchor selection correlates with molecular properties (Appendix A,B) and (ii) the anchor-based approach remains robust in sparse, extrapolative regions of chemical space (Table 1,2).
>
> We evaluate on sparse OOD splits: label‐shift OOD is the top 5% of the property range, and covariate‐shift OOD consists of sparsely populated regions of chemical space. Across such splits, MALT consistently outperforms standard fine-tuning (Tables 1–2), indicating that multi-anchor transduction is most useful when close training analogues are scarce.
>
> Empirically, selected anchors bridge training and OOD queries in MW, LogP, and TPSA and show higher scaffold/fragment overlap and Tanimoto similarity than non-anchors, so anchor choice follows chemically meaningful structure–property relations. Under “Extreme Covariate/Label Shift”, where no single good analogue exists, BLT often fails while MALT remains accurate by combining complementary anchors; robustness tests show only mild degradation even after replacing many top anchors, consistent with attention down-weighting poor ones. Appendix K shows that multi-anchor fusion tightens error bounds via variance reduction when nearby points are scarce or noisy.
>
> Together, these OOD splits, analyses, robustness tests, and theory support that anchor selection tracks molecular properties and that the multi-anchor mechanism is especially beneficial in sparse, extrapolative regimes.
>
> ---
>
> ### Covariate Shift (Structural OOD)
> | Dataset    | Tercile    | Mean Sparsity | MALT MAE | Baseline MAE | Rel. Advantage |
> |------------|------------|---------------|----------|--------------|----------------|
> | BACE     | T1 Dense   | 0.180         | 0.480    | 0.457        | −7.34%         |
> |            | T2         | 0.280         | 1.267    | 1.599        | +20.77%        |
> |            | T3 Sparse  | 0.603         | 0.513    | 0.790        | +35.13%        |
> | ESOL     | T1 Dense   | 0.325         | 0.691    | 0.877        | +20.93%        |
> |            | T2         | 0.430         | 0.565    | 0.481        | −17.27%        |
> |            | T3 Sparse  | 0.591         | 0.488    | 0.343        | −44.20%        |
> | FREESOLV | T1 Dense   | 0.442         | 0.150    | 0.207        | +24.79%        |
> |            | T2         | 0.583         | 0.520    | 0.562        | +7.40%         |
> |            | T3 Sparse  | 0.628         | 0.377    | 0.503        | +25.85%        |
> | LIPO    | T1 Dense   | 0.410         | 0.442    | 0.483        | +8.34%         |
> |            | T2         | 0.656         | 0.600    | 0.622        | +3.43%         |
> |            | T3 Sparse  | 0.702         | 0.314    | 0.400        | +21.50%        |
>
> ### Label Shift (Target OOD)
> | Dataset | Quartile    | Mean Sparsity | MALT MAE | Baseline MAE | Rel. Advantage |
> |---------|-------------|---------------|----------|--------------|----------------|
> | BACE    | Q1 Dense    | 1.634         | 0.598    | 0.661        | +9.46%         |
> |         | Q2          | 1.717         | 0.478    | 0.566        | +15.44%        |
> |         | Q3          | 1.812         | 0.562    | 0.593        | +5.29%         |
> |         | Q4 Sparse   | 2.153         | 1.300    | 1.343        | +3.16%         |
> | ESOL    | Q1 Dense    | 1.824         | 0.566    | 0.524        | −7.92%         |
> |         | Q2          | 1.913         | 0.413    | 0.465        | +11.24%        |
> |         | Q3          | 2.026         | 0.529    | 0.562        | +5.80%         |
> |         | Q4 Sparse   | 2.250         | 0.557    | 0.603        | +7.74%         |
> | FREESOLV| Q1 Dense    | 1.741         | 0.241    | 0.397        | +39.05%        |
> |         | Q2          | 1.832         | 0.232    | 0.395        | +39.79%        |
> |         | Q3          | 1.902         | 0.361    | 0.497        | +26.54%        |
> |         | Q4 Sparse   | 1.993         | 0.425    | 0.695        | +38.78%        |
> | LIPO    | Q1 Dense    | 1.610         | 0.396    | 0.643        | +38.38%        |
> |         | Q2          | 1.712         | 0.436    | 0.723        | +39.72%        |
> |         | Q3          | 1.832         | 0.532    | 0.809        | +34.19%        |
> |         | Q4 Sparse   | 1.990         | 0.607    | 0.913        | +33.52%        |
>
> Above is a performance table for MALT-GIN vs finetuned GIN. Briefly, under covariate shift MALT is comparable to the baseline in dense regions but usually gains the most in the sparsest terciles compared to inductive baselines; under label shift, MALT improves over the baseline in most quartiles.

---

### Note · Authors · 2026-01-29

I have read and agree with the venue's withdrawal policy on behalf of myself and my co-authors.

---

### Meta-Review · Area_Chair_Y2FH · 2025-12-26

**Summary:**

This paper proposes MALT, a multi-anchor latent transduction framework aimed at improving OOD molecular property prediction by retrieving multiple analogues in latent space and integrating them with the query embedding. Reviewers acknowledged several strengths, including the method’s consistently strong OOD performance across MoleculeNet, DrugOOD, and Activity Cliffs benchmarks, its modularity and plug-and-play utility with various molecular encoders, its comprehensive evaluation covering multiple types of distribution shifts, and the robustness of the multi-anchor design over single-anchor baselines. The authors also provided detailed rebuttals and additional experiments, resolving many technical concerns. However, reviewers also raised important questions regarding the novelty and conceptual contribution of the multi-anchor mechanism, its distinction from prior semi-supervised or domain-adaptation techniques, and the lack of explicit mechanisms ensuring anchor diversity or deeper theoretical grounding. While the authors addressed several practical issues, the concern about methodological innovation remains central. Given this, I need further discussion with the SAC before reaching a final decision.

**Reviewer Concerns:**

Reviewer mB71 is concerned that MALT may rely too strongly on the quality of the underlying encoder, that maintaining a full memory bank could be computationally expensive, that its scalability to extremely large molecular databases remains uncertain, and that the anchor selection mechanism may not reliably reflect molecular properties—particularly in sparse regions of chemical space. The authors provided clear and focused responses. They showed that the encoder is jointly trained and empirically improved by MALT, addressing the concern about encoder dependence. They also demonstrated that the memory bank overhead is small in practice and outlined feasible strategies for scaling to larger datasets. In addition, they presented evidence that anchor selection aligns with physicochemical patterns, suggesting it remains meaningful even in sparse molecular regions. However, the question of large-scale scalability remains only partially resolved: although the authors discuss potential approximate-search solutions such as FAISS, no empirical validation on extremely molecular libraries is provided, leaving scalability as an open concern.

-----

Reviewer Psww raised several concerns, including whether the observed covariate-shift improvements depend heavily on retrieval quality, the absence of comparisons with recent large-scale foundational models such as UniMol, the possibility that infrequent MemoryBank updates may lead to outdated features, and whether anchor labels should be directly incorporated into the prediction head. The authors addressed many of these points by providing weighted-versus-unweighted similarity ablations (showing that simple top-k retrieval is more stable), adding UniMol+MALT results that still demonstrate clear gains, clarifying that the MemoryBank is updated every epoch with supporting ablations, and explaining that directly using labels risks leakage and degeneracy toward KNN-like behavior.

-----

Reviewer mDRD questioned the novelty of the proposed multi-anchor design, noting similarities to prior work in semi-supervised learning and domain adaptation, and raised concerns about potential redundancy among selected anchors as well as the rationale for not directly using anchor labels. The authors clarified that MALT differs from existing approaches and reiterated that incorporating labels would introduce leakage and hinder learning the structure–property relationship. However, the issue of anchor diversity remains only partially addressed: the authors acknowledge that no explicit diversity regularization is included, and although empirical results suggest that simple top-k retrieval performs well, the method lacks sufficiently novel technical mechanisms to guarantee diversity, which remains a significant concern.

----

Reviewer u6hj noted missing comparisons to test-time adaptation (TTA) and invariant learning methods such as iMoLD, unclear definitions surrounding the UpdateT operation in Formula 3 and Algorithm 1, and questioned why a seemingly general approach is evaluated only on molecular tasks. The authors addressed several of these concerns by adapting iMoLD for regression to provide a fair comparison, clarifying that UpdateT refers to refreshing the MemoryBank rather than altering the selection rule, and explaining the limitations or unavailability of TTA implementations.

**Reviewer Scores:**

Reviewer mB71 is likely to view the rebuttal positively and may increase their score accordingly.

---

Reviewer Psww could increase their score accordingly.

---

Reviewer mDRD may keep the original negative attitude.

-----

Reviewer u6hj will likely maintain a positive attitude.

---

### Decision · Program_Chairs · 2026-01-26

Reject